# Breaking the cell wall for efficient DNA delivery to diatoms

E. J. L. Walker [1], M. Pampuch [1], L. Deng[2], Y. Li[2], G. Tran[1], T. Mock [2] & B. J. Karas [1] ✉

Diatoms produce 20% of the world's fixed organic carbon yet remain under-utilized as cell factories due to limited genetic engineering tools. Here, we present optimized electroporation and polyethylene glycol (PEG) transformation methods for the model diatom *Phaeodactylum tricornutum*, enabling delivery of DNA and Cas9 ribonucleoprotein complexes with high efficiency. Transformants are recovered with as little as 1 ng of DNA, and linear or circular episomes as large as 55.6 kb are successfully introduced. The optimized electroporation protocol also reveals an unexpected capability: episomes can be assembled directly in the algal cell through non-homologous or homology-driven repair mechanisms, a process we term *diatom* in vivo *assembly* (DIVA). In addition, the PEG approach is adapted to successfully transform *Thalassiosira pseudonana*, demonstrating the applicability of our methods for engineering other diatom species. These tools could be used to accelerate diatom synthetic biology projects and, therefore, the development of sustainable technologies.

Climate change, population expansion, and soil depletion are among several challenges that existentially threaten current agricultural practices. Due to their comparatively rapid growth and higher photosynthetic efficiency, unicellular algae have been proposed as a sustainable agricultural alternative to traditional crops since the 1950s[1]. Diatoms (Bacillariophyta) are of particular interest for this purpose as they are widely distributed across the world's oceans[2], where they are one of the most photosynthetically productive taxa – accounting for an estimated 20% of total fixed organic carbon annually[3].

*Phaeodactylum tricornutum* is considered to be one of the most characterized model diatom species, having a fully sequenced telomere-to-telomere reference genome[4] in addition to molecular tools for DNA transformation[5–8], CRISPR-based genome engineering[9], protein expression[10], cloning of the whole organelle genomes[11,12], and more. These characteristics have made *P. tricornutum* an emerging chassis for diatom biotechnology[13] and the subject of a whole-genome synthesis effort[14]. The latter, termed the Pt-syn 1.0 project, has brought together an international consortium of phycologists with the aim of replacing all the native chromosomes in the nuclear, chloroplast, and mitochondrial genomes with synthetic counterparts. To actualize this, and to discover the full potential of diatoms as cell factories, a pipeline for efficiently and rapidly testing different DNA constructs in *P. tricornutum* must be established.

Existing methods for delivering DNA to *P. tricornutum* include biolistic bombardment[8], electroporation[5,7], polyethylene glycol (PEG) transformation[6], and bacterial conjugation[6]. Biolistics was the first method developed for transforming *P. tricornutum* and has been used to integrate non-replicative DNA into the nuclear genome[8,15], with reported efficiencies of 1 to 100 transformants per $10^8$ cells. Though reasonably efficient, this method requires the use of expensive equipment (i.e., a biolistic particle delivery system) that many labs do not have access to; comparatively, conjugation, electroporation, and PEG transformation can all be conducted with more standard equipment.

Among these, conjugation is the most efficient transformation method for delivering episomes (4 transformants per $10^4$ cells)[6], but also the most mutation prone and laborious to execute[6,9]. Conjugation requires the creation and maintenance of a donor strain of *Escherichia*

[1]Department of Biochemistry, Schulich School of Dentistry and Medicine, London, ON, Canada. [2]School of Environmental Sciences, University of East Anglia, Norwich Research Park, Norwich, UK. ✉e-mail: bkaras@uwo.ca

*coli*, which must contain a suitable broad-host-range conjugative plasmid in addition to the episome of interest; complications can arise if these two constructs have incompatible replication machinery[16], if the episome is maintained at medium to high copy number[6], or if it contains elements that are toxic when expressed in *E. coli* (e.g., endonucleases)[9]. These issues can lead to episomal rearrangements or loss over time. As a result, donor strains often require repeated reisolation and plasmid sequencing to confirm construct stability – steps that can add days to weeks, and in some cases months, to a workflow.

The PEG and electroporation transformation methods offer a more direct path to episomal delivery, but each has limitations. The delivery of episomes through PEG transformation remains unutilized due to its low efficiency ($\leq 1$ transformant per $10^8$ cells) and high variability between experiments[6]. Although PEG-mediated transformation has not yet been demonstrated in other diatom species, it is routinely applied in the microalga *Chlamydomonas reinhardtii*, where cell-wall removal followed by PEG-mediated agitation with DNA and glass beads enables efficient nuclear transformation[17]. Electroporation has a higher transformation efficiency (2.6 to 4.5 transformants per $10^5$ cells)[5,7], but has not received wide adoption due to variability in efficiency and issues with reproducibility. Thus, we sought to optimize the PEG and electroporation methods for episome delivery to *P. tricornutum* with the hopes of developing a more rapid alternative to bacterial conjugation.

In this paper we describe optimized electroporation and PEG transformation protocols for engineering *P. tricornutum*. These methods are both faster and simpler to execute than bacterial conjugation, as well as more efficient than previously described protocols. We also demonstrate that it is possible to assemble episomes directly in the algal cell through non-homologous repair (NHR) and homology-directed repair (HDR) pathways, opening up avenues for exploring alternative assembly strategies that do not rely on traditional *E. coli* or *S. cerevisiae* cloning. Both protocols are suitable for rapid high throughput testing of several DNA constructs simultaneously and, due to their efficiency with low DNA amounts, could be used for the delivery of commercially synthesized episomes directly into the algal cell. Electroporation can also be used for the efficient delivery of Cas9 ribonucleoprotein (RNP) complexes, enabling DNA-free genetic engineering. The protoplasting techniques we establish in *P. tricornutum* are also used to successfully engineer *Thalassiosira pseudonana* cells via PEG transformation, demonstrating that these methods can be applied to other diatom species. These tools will be valuable as the Pt-syn 1.0 project begins to build and test different synthetic chromosome designs.

## Results

### Electroporation efficiency increases with enzymatic treatment of cells

Our initial attempts at electroporation using a previously described protocol[18] that was adapted from the methods of Zhang and Hu[5] yielded no transformants. After several failed attempts, we discovered that adjusting the electroporator capacitance from 25 to 50 μF yielded transformants for both episomes and marker cassettes, though efficiency was highly variable between experiments (Supplementary Data 1, Exp. 1 to 81; Supplementary Fig. 1). We used this method to deliver a linear 11 kb PCR-amplified episome, pPtGE31_ΔPtR (Addgene ID: 236260), to assess whether the construct would integrate into the genome, persist as a linear episome, or be converted into a circular episome within the *P. tricornutum* nucleus. We were able to recover DNA from these algal transformants in EPI300 *E. coli*, indicating that the linear episome had been circularized in the algal nucleus following electroporation (Supplementary Fig. 2). Whole-plasmid sequencing of four *E. coli* colonies confirmed that circularization had occurred via fusion of the construct's termini.

When this electroporation protocol was repeated using cells that had reached the late stationary phase of growth, we obtained up to 272 transformants per $10^8$ cells with the same 11 kb PCR-amplified episome (Supplementary Data 1, Exp. 82 to 89). This represented a nearly 20-fold increase in efficiency when compared to transformations with early-log phase cultures (Exp. 80 and 81). Stationary phase cultures demonstrated noticeable differences in cell phenotype when compared to early-log phase cultures, with there being a higher abundance protoplasts (i.e., diatoms lacking a cell wall) in the older cultures. *P. tricornutum* cells can appear in three different morphotypes – fusiform, oval, and triradiate – and can switch morphotypes in response to environmental stimuli[19,20]. All morphotypes can generate protoplasts if the cell wall is sufficiently weakened. Ultimately, this observation led us to hypothesize that cell morphotype and cell wall integrity may influence electroporation transformation efficiency. Notably, a spheroplasted cell has only partial cell wall degradation and may not appear phenotypically different from a wild-type cell, whereas a fully protoplasted cell lacks a cell wall entirely and adopts a characteristically spherical morphology – provided it does not burst from osmotic or mechanical stressors.

Cell morphotype cannot be readily controlled in *P. tricornutum* cultures as this species displays a high degree of phenotypic plasticity; however, agar-plated cultures of CCAP 1055/1 tend to have a higher proportion of oval-type cells (~20–30% oval, ~70–80% fusiform) compared to liquid cultures (<10% oval, >90% fusiform). Thus, to test this hypothesis, we spheroplasted early-log phase *P. tricornutum* cells harvested from agar plates. Plate-derived cells were spheroplasted using the enzyme alcalase, a serine protease that has been shown to degrade the proteinaceous components of the *P. tricornutum* cell wall[21]. We observed that 100 μl of alcalase at an activity of approximately 3 Anson units per ml (i.e., 300 mAnson units) was enough to fully protoplast $3 \times 10^8$ plate-derived cells when resuspended in 375 mM D-sorbitol (Fig. 1a, b). The protoplasted cells were very fragile and prone to rupture from osmotic pressure or physical damage; treatment of $3 \times 10^8$ cells with 10 μl of alcalase (i.e., 30 mAnson units) was enough to protoplast some cells, whilst leaving others with partially degraded cell walls (i.e., spheroplasts; Supplementary Fig. 3).

Plate-derived cells ($3 \times 10^8$) were treated with 0.3 mAnson units, 3 mAnson units, and 30 mAnson units of alcalase prior to the washing steps of the electroporation protocol. Transformation efficiency increased in all cases (Fig. 1c) when electroporating 500 ng of the same 11 kb PCR-amplified episome as before (Fig. 1d), with the 3 mAnson treatment demonstrating the highest increase in efficiency. Based on this result, we went forward with using 1 mAnson unit of alcalase per $10^8$ cells when spheroplasting *P. tricornutum* for electroporation.

### Electroporation efficiency of cells grown in liquid and solid media

We tested the electroporation efficiency of *P. tricornutum* cells grown in liquid media and on agar plates, with and without alcalase treatment across five biological replicates (Fig. 1e) using 500 ng of the same PCR-amplified 11-kb episome as before. Though transformation efficiency varied between experiments, plate-derived alcalase-treated cells were significantly more efficient (~1.2 transformants per $10^4$ cells) than untreated plate-derived cells (~6.0 transformants per $10^7$ cells), untreated liquid-derived cells (~4.2 transformants per $10^7$ cells), and treated liquid-derived cells (~1.6 transformants per $10^6$ cells, Supplementary Table 1). Statistical significance was determined using a Kruskal-Wallis test ($p = 0.0094$) followed by pairwise Wilcoxon tests with Bonferroni correction. Plate-derived, alcalase-treated cells yielded significantly more colonies than all other groups ($p < 0.05$), whereas no significant differences were observed among the remaining comparisons.

Based on this, all proceeding electroporations were conducted with cells derived from plates and spheroplasted with 1 mAnson unit

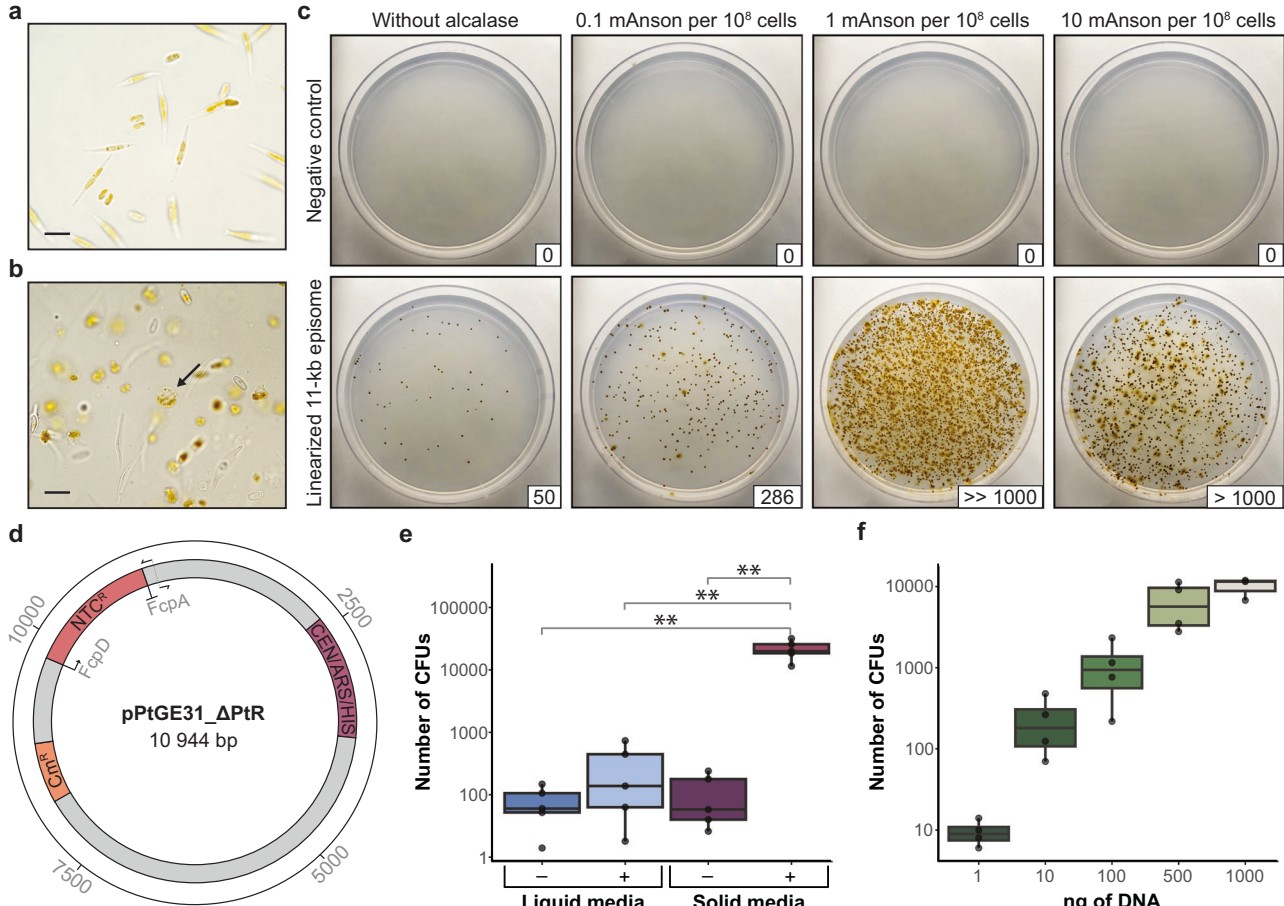

**Fig. 1 | Spheroplasting *P. tricornutum* cells increases transformation efficiency.** **a** Plate-derived *P. tricornutum* cells pre- and **b** post-treatment with 100 mAnson units of alcalase per $10^8$ cells. The arrow indicates a completely protoplasted cell, and the scale bars represent 10 μm. **c** Comparison of agar plate-derived *P. tricornutum* cells treated with differing amounts of alcalase prior to electroporation. Cells were either electroporated with no DNA (negative control) or 500 ng of the 11 kb episome pPtGE31_ΔPtR, which was obtained through PCR amplification. The number of colony-forming units (CFUs) is depicted in the bottom right corner of each transformation plate. Following electroporation, half of the total reaction was plated on ¼-salt L1 plates supplemented with 100 μg/ml nourseothricin. **d** A plasmid map of pPtGE31_ΔPtR, the episome used for transformations in panels **c**, **e**, and **f** Primers depict where the plasmid was linearized during PCR-amplification. This plasmid contains a yeast centromere (CEN) and origin of replication (ARS) sequence, which enables episome replication and stability in *P. tricornutum*[6], as well as a bacterial chloramphenicol resistance marker (Cm^R) and an algal nourseothricin resistance marker (Ntc^R) driven by the FcpD/FcpA promoter-terminator pair[9]. **e** A boxplot demonstrating transformation efficiency for cells cultured in liquid or solid media with (+) or without (-) alcalase treatment (1 mAnson per $10^8$ cells) prior to electroporation with 500 ng of PCR amplified pPtGE31_ΔPtR. Five biological replicates were performed. Asterisks denote pairs that are significantly different in terms of efficiency following a two-sided statistical analysis (Kruskal-Wallis test, $p = 0.0094$; Bonferroni-corrected Wilcoxon rank-sum tests: $p = 0.048$ for all comparisons to solid media-treated cells). **f** A boxplot demonstrating the number of *P. tricornutum* CFUs obtained with different nanograms of DNA during electroporation. The cells used in these experiments were derived from agar plates and treated with 1 mAnson of alcalase per $10^8$ cells prior to electroporation; four biological replicates were performed. Box plots show the median (centre line), interquartile range (box, 25th–75th percentile), and whiskers extending to 1.5X the interquartile range. Points represent individual biological replicates.

of alcalase per $10^8$ cells. This optimized method was compared with the original and adjusted electroporation protocols to demonstrate the impact that changing the capacitance parameter and spheroplasting have on electroporation efficiency (Supplementary Fig. 1).

### Transformation efficiency with varying amounts of DNA

Our prior transformations were all conducted with ~500 ng of a PCR-amplified 11-kb plasmid, pPtGE31_ΔPtR (Fig. 1d). We were interested to see how transformation efficiency varied when using differing amounts of this construct, and most importantly, what the minimum amount of DNA for transformation is. We electroporated as little as 1 ng and up to 1 μg of PCR-amplified pPtGE31_ΔPtR across four biological replicates (Fig. 1f, data summarized in Supplementary Table 2). In all but one experiment, we were able to recover transformants when using just 1 ng of DNA during electroporation. Increasing the amount of DNA to 10 ng routinely gave rise to hundreds of transformants, with

there being an average of 372 CFUs per reaction. Interestingly, there was only a small change in transformation efficiency when the amount of DNA was doubled from 500 ng to 1 μg, suggesting that 500 ng of linear pPtGE31_ΔPtR was enough to nearly saturate the electroporation reaction.

### Transforming linear and circular episomes to the algal nucleus

Our past explorations had relied upon the use of linear PCR-amplified DNA with blunt, non-phosphorylated termini. To test the efficiency of different structural forms of DNA, we prepared other variations of pPtGE31_ΔPtR (Fig. 2a). The termini of the linear constructs varied in position along the episome and/or the properties of the termini (i.e., blunt or sticky ends, with or without 5′ phosphorylation). These constructs were adjusted to be equimolar prior to conducting electroporation in *P. tricornutum*. We found that there were nearly double the amount of CFUs when transforming any structural form of linear DNA compared to the circular episome (Fig. 2b).

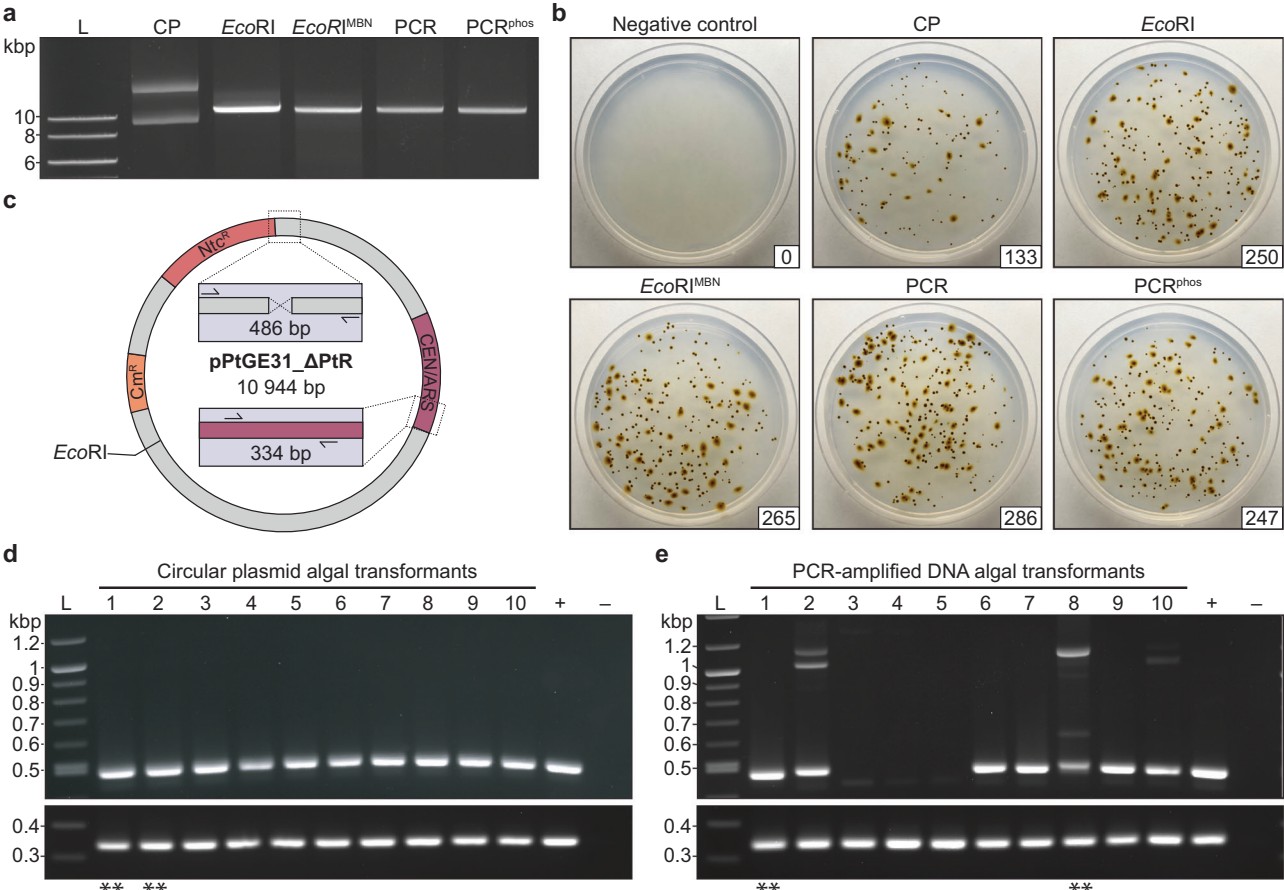

**Fig. 2 | Comparing the transformation efficiency of linear, circularized, and phosphorylated DNA. a** An agarose gel depicting different preparations of the episome pPtGE31_ ΔPtR. CP: circular plasmid, *Eco*RI: restriction enzyme digested plasmid, *Eco*RI^(MBN): restriction enzyme digested plasmid with additional mung bean nuclease treatment to remove sticky ends, PCR: PCR-amplified linear plasmid DNA, and PCR^(phos): PCR-amplified linear plasmid DNA generated with phosphorylated primers. **b** Comparison of *P. tricornutum* transformants electroporated with different preparations of pPtGE31_ ΔPtR. Following electroporation, one-tenth of the total reaction was plated on ¼-salt L1 plates supplemented with 100 μg/ml nourseothricin. Plates are labeled according to the type of DNA that was transformed, as described above. The number of CFUs is shown in the bottom right corner of each

plate. **c** A plasmid map depicting the regions that were screened when assessing passaged algal transformants. The 486 bp junction spans the region where pPtGE31_ ΔPtR was split for PCR amplification, whereas the 334 bp junction spans an intact region of the plasmid backbone. **d** Screening 10 algal colonies that had been transformed with circular pPtGE31_ ΔPtR or **e** PCR-amplified pPtGE31_ ΔPtR. The primer pairs span the termini region (expected size of 463 bp, top panel) and the CEN/ARS region of the episome backbone (expected size of 334 bp, bottom panel). Primer pairs are listed in Supplementary Data 2. Dilute pPtGE31_ ΔPtR and genomic *P. tricornutum* DNA were used for the positive and negative controls, respectively. Double asterisks (**) indicate the algal episomes that would be recovered in *E. coli* and sent for whole plasmid sequencing.

We had also demonstrated that electroporated linear pPtGE31_ ΔPtR gets circularized in the *P. tricornutum* nucleus, but we had not yet performed comprehensive screening to determine if any indels were accruing during this repair process. To explore what occurs during termini fusion, ten colonies were repatched from the transformations with circular and PCR-amplified pPtGE31_ ΔPtR for additional screening. Colonies were screened using a set of primers that spans where the termini are expected to recombine for the PCR-amplified construct, as well as a region of the episome backbone (Fig. 2c). Colonies transformed with circular DNA did not demonstrate any unexpected insertions or deletions in the screened regions (Fig. 2d), as expected. Whole plasmid sequencing of circular colonies 1 and 2 confirmed that the episomes had not accrued any mutations or indels during transformation.

Colonies transformed with PCR-amplified DNA appeared to have indels in the region where termini fusion occurs but not in the intact backbone region (Fig. 2e). We conducted Sanger sequencing of the termini region for colonies 1, 2, and 6 to 10, which confirmed that indels of varying sizes had accrued in this region for every colony

screened (Supplementary Fig. 4). The absence of the 486 bp band in lanes 3, 4, and 5 is most likely due to DNA degradation or deletions occurring at the episomal termini, either before (e.g., exonuclease activity) or during the circularization process. Whole plasmid sequencing of linear colonies 1 and 8 further confirmed the presence of indels in the termini region where the episome was being fused together, whereas all other regions of the episome had remained unchanged. Interestingly, the ~500-bp insertions in colonies 2, 8, and 10 were not complementary to any regions of the episome or algal genome. When these insertions were queried using BLASTn, they aligned to various regions of the salmon and/or trout genome accessions.

Single-stranded salmon sperm (ssss) DNA is used as a carrier during electroporation, and to our surprise, we had discovered that it could be captured during NHR of the episome termini. We attempted electroporation without ssssDNA to see if this would impact transformation efficiency, and it resulted in a 15-fold reduction in the number of CFUs (Supplementary Fig. 5), highlighting its importance as an additive for efficient transformation.

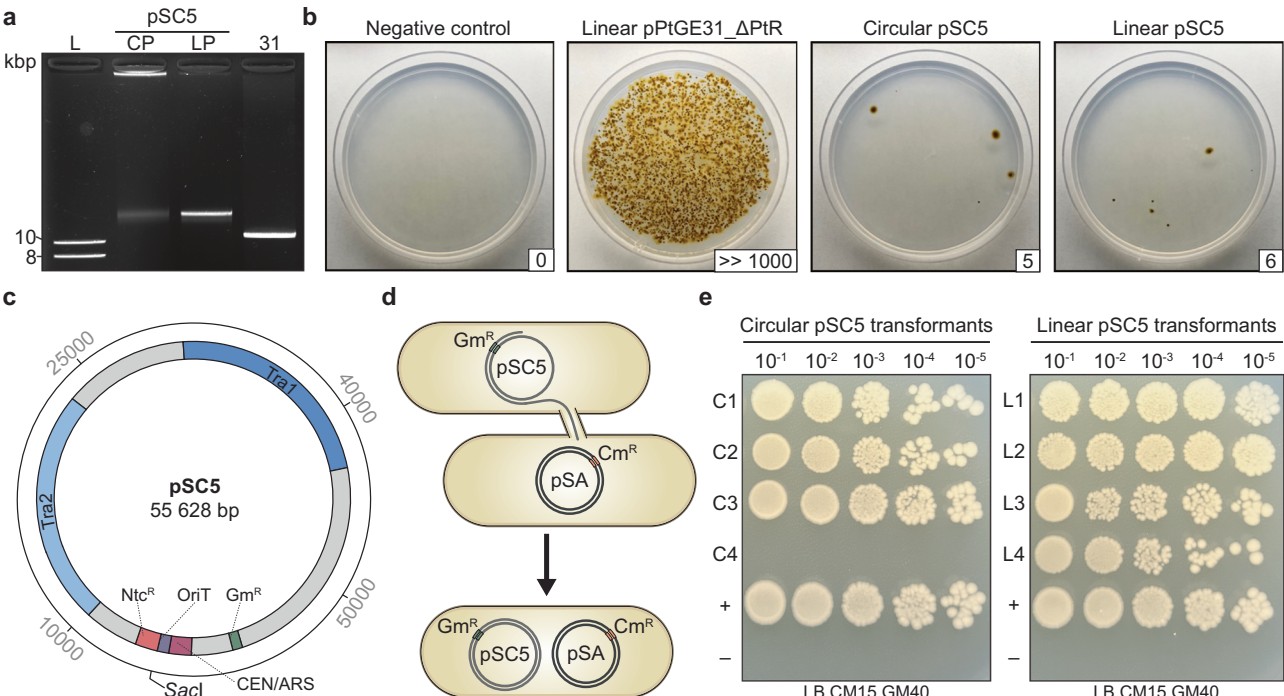

**Fig. 3 | Electroporation and recovery of a 55.6 kb conjugative plasmid. a** An agarose gel depicting the circular pSC5 (CP), linearized pSC5 (LP), and PCR-amplified pPtGE31_ΔPtR (31). **b** Electroporation of circular or linear variants of pSC5 and PCR-amplified pPtGE31_ΔPtR into *P. tricornutum* cells. Following electroporation, half of the total reaction was plated on ¼-salt L1 plates supplemented with 100 μg/ml nourseothricin. Plates are labeled according to the type of DNA that was transformed, as described above. The number of CFUs is shown in the bottom right corner of each plate. **c** A plasmid map of pSC5. The transfer regions (Tra1 and Tra2) encode essential conjugative proteins. The origin of transfer (OriT) serves as a recognition sequence for the conjugative protein apparatus, facilitating the transfer of pSC5 during conjugation. The plasmid contains selection markers for bacteria (Gm^R, gentamycin resistance) and *P. tricornutum* (Ntc^R, nourseothricin resistance)

as well as a yeast centromere (CEN) and replication origin (ARS), which enable episome stability and replication in *P. tricornutum*. **d** A donor strain of *E. coli* harboring pSC5, which confers gentamycin resistance, conjugates to a recipient strain of *E. coli* harboring pSAP, which confers chloramphenicol resistance. Successful conjugation leads to the creation of a transconjugant strain that is gentamycin and chloramphenicol-resistant. **e** Testing the conjugation efficiency of *E. coli* strains harboring episomes that had been recovered from *P. tricornutum* transformants electroporated with circular (C1 to C4) or linear (L1 to L4) pSC5. Strains harboring pSC5 or pSAP were used as the positive and negative controls, respectively. Dilutions of $10^{-1}$ to $10^{-5}$ were spot plated on LB plates supplemented with 15 μg/ml chloramphenicol and 40 μg/ml gentamicin.

## Evaluation of genomic integration

We performed a plasmid/episome loss experiment by serially passaging four pPtGE31_ΔPtR transformants in non-selective media to check for genomic integration. The transformants chosen correspond to algal colonies 1 and 2 in Figs. 2d and 1 and 8 in Fig. 2e. After seven passages in liquid media with or without nourseothricin selection, serial dilutions of the cultures were then plated onto solid non-selective media to obtain single colonies. For each transformant, 100 single colonies were then re-patched onto plates supplemented with nourseothricin. If genomic integration of the episome had occurred, we would expect a high proportion of the single colonies to retain resistance. Conversely, if the episomes are maintained extra chromosomally, they should be lost over successive generations without selective pressure, and thus only a low proportion of colonies would retain resistance. As expected, only 2–7% of screened colonies retained resistance (Supplementary Table 3), demonstrating that genomic integration had not occurred in the assayed transformants.

## Electroporation and recovery of a large plasmid

It has been demonstrated in other organisms that as episome size increases, electroporation efficiency decreases[22,23]. We sought to test the limits of electroporation in *P. tricornutum* by attempting transformation with pSC5, a 55.6 kb broad-host-range conjugative episome (Fig. 3c, Addgene ID: 188602)[24].

Equimolar amounts of circular and linear pSC5 were prepared along with PCR-amplified pPtGE31_ΔPtR (Fig. 3a). Electroporation with either form of pSC5 gave rise to transformants, though the number of

CFUs was orders of magnitude lower than electroporation with a linear 11 kb episome (Fig. 3b), as anticipated. To determine if the complete 55.6 kb episome had been successfully electroporated into *P. tricornutum*, we passaged four colonies from each of the transformations with circular and linear pSC5. DNA was isolated from the algal colonies and electroporated into *E. coli*. Here, if the *E. coli* transformants contain an intact version of pSC5, the episome will facilitate conjugation and self-mobilization into a recipient strain harboring pSAP[12] (Addgene ID: 206429), a 10 kb plasmid that contains a chloramphenicol resistance marker (Fig. 3d). Successful transconjugants will harbor pSC5 and pSAP, therefore carrying resistance to both gentamycin and chloramphenicol.

We performed conjugation using a pool of *E. coli* colonies for each transformation (Fig. 3e; C1–C4 are derived from circular pSC5 algal transformants, L1 to L4 are derived from linear pSC5 algal transformants). Conjugation was successful for three out of four (circular pSC5) and four out of four (linear pSC5) of the *E. coli* pools tested, demonstrating that intact pSC5 had been successfully electroporated and maintained in the majority of screened *P. tricornutum* transformants (Fig. 3e).

## Electroporation and recombination of multiple DNA fragments
**Assembly via a non-homologous repair pathway.** The surprising discovery of salmon DNA inserted into the termini region of PCR-amplified pPtGE31_ΔPtR led us to hypothesize that *P. tricornutum* could assemble multiple DNA fragments into a single episome through NHR. To test this, we created an episome, pPtGE31_ShBle (Addgene ID:

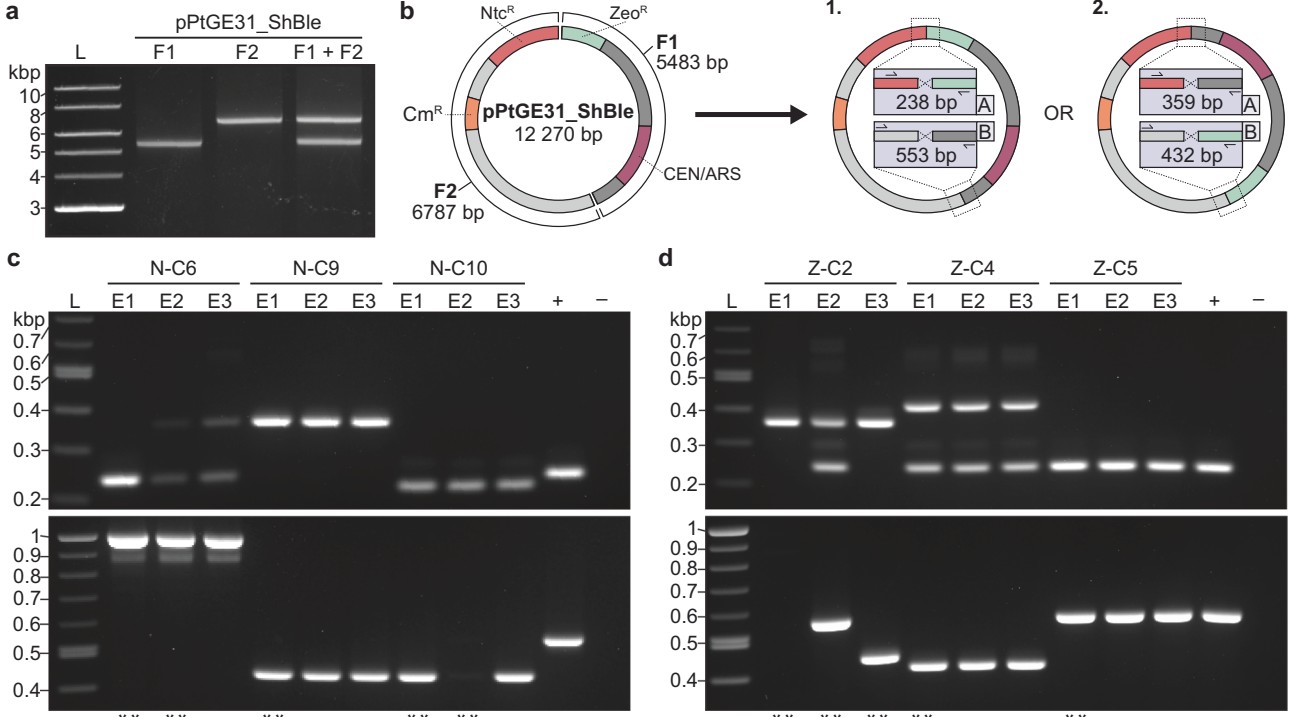

**Fig. 4 | Electroporation and assembly of two non-overlapping fragments in *P. tricornutum*. a** The episome pPtGE31_ShBle was amplified as two fragments (F1: fragment 1, F2: fragment 2). The fragments were adjusted to be equimolar before combining them into an assembly mixture (F1 + F2). **b** A plasmid map of pPtGE31_ShBle and a depiction of the different possible orientations post-assembly. The episome contains algal selection markers for Ntc^R and Zeo^R, a bacterial Cm^R selection marker, as well as a yeast CEN and ARS sequences. F1 and F2 do not share any overlapping sequences and thus can assemble in two different orientations. Primer sets were designed to span the two junctions, A and B, where recombination is expected to occur. The amplicons will vary in size depending on the orientation of the fragments. **c** Screening three *E. coli* transformants per algal colony using primers spanning junctions A (top panel, expected size of 238 or 359 bp) and B (bottom panel, expected size of 553 or 432 bp). DNA was isolated from algal transformants that were electroporated with F1 + F2 and selected initially with nourseothricin or **d** zeocin. Dilute pPtGE31_ ShBle and water were used for the positive and negative controls, respectively. Double asterisks (\*\*) indicate the *E. coli* colonies that were sent for whole plasmid sequencing.

236261), that contains the NAT and ShBle markers conferring resistance to nourseothricin and zeocin, respectively. The episome pPtGE31_ShBle was then amplified as two non-overlapping fragments, each containing one of the algal selective markers. The fragments were adjusted to be equimolar prior to electroporation into *P. tricornutum* (Fig. 4a).

Cells transformed with both fragments simultaneously were plated on nourseothricin- or zeocin-containing plates initially (Supplementary Fig. 6). Out of the 116 colonies passaged post-electroporation, 94 demonstrated double resistance to both nourseothricin and zeocin (Supplementary Table 4). Twenty transformants, ten derived from each initial transformation plate, were passaged onto double-selection media, then DNA isolation and PCR screens were performed. These colonies are denoted with an N- or Z- to depict which initial selection plate they were derived from (N: nourseothricin, Z: zeocin).

Two orientations are expected when the fragments amplified from pPtGE31_ShBle are re-assembled (Fig. 4b). We designed a PCR screen that would not only assess if both fragments had been recombined, but could also determine the orientation of the fragments. When screening algal transformants, we only used the primer set spanning junction B as junction A contains primers that bind to the FcpD promoter and FcpA terminator, which are present in both the episome and algal genome. Of the twenty colonies screened, nine had amplification across junction B, with only five colonies demonstrating the expected amplicon sizes (Supplementary Fig. 7). The other four colonies either demonstrated multiple amplicons, some of which were the expected sizes, and/or amplicons that were larger than expected.

Based on this initial screen, DNA from colonies N-C6, N-C9, N-C10, Z-C2, Z-C4, and Z-C5 was transformed into *E. coli* for further screening.

Three *E. coli* transformants were assessed per algal colony using the primer sets spanning junctions A and B (Fig. 4c, d). Interestingly, the *E. coli* transformants derived from algal colonies N-C6, N-C10, and Z-C2 demonstrated different banding patterns, suggesting that these algal colonies likely had a pool of assembled episomes. All *E. coli* transformants from algal colony N-C9 depicted the expected banding pattern for orientation 2, whereas all *E. coli* transformants from algal colony Z-C5 depicted the expected banding pattern for orientation 1. DNA from *E. coli* colonies N-C6-E1/E2, N-C9-E1, N-C10-E1/E2, Z-C2-E1/E2/E2, Z-C4-E1, and Z-C5-E1 were sent for whole-plasmid sequencing to validate if assembly had occurred.

The sequencing results demonstrated that recombination of fragments 1 and 2 had occurred in all assessed *P. tricornutum* transformants (Supplementary Fig. 8). In the majority of assembled episomes, multiple partial copies of fragments 1 and 2 had recombined. The most excessive recombination appeared *E. coli* colony Z-C2-E2, which harbored a 41.5 kb episome containing eight partial or whole copies of both fragments 1 and 2. Only *E. coli* colonies N-C9-E1, N-C10-E1, Z-C2-E3, and Z-C5-E1 contained the expected assembled episomes. Sequencing of *E. coli* transformants derived from the same algal colony revealed that algal colonies N-C6, N-C10, and Z-C2 possessed a pool of assembled episomes, as we had hypothesized.

**Assembly via a homology-directed repair pathway.** Though it is possible to assemble episomes via the NHR pathway, screening and sequencing demonstrated that the majority of *P. tricornutum*

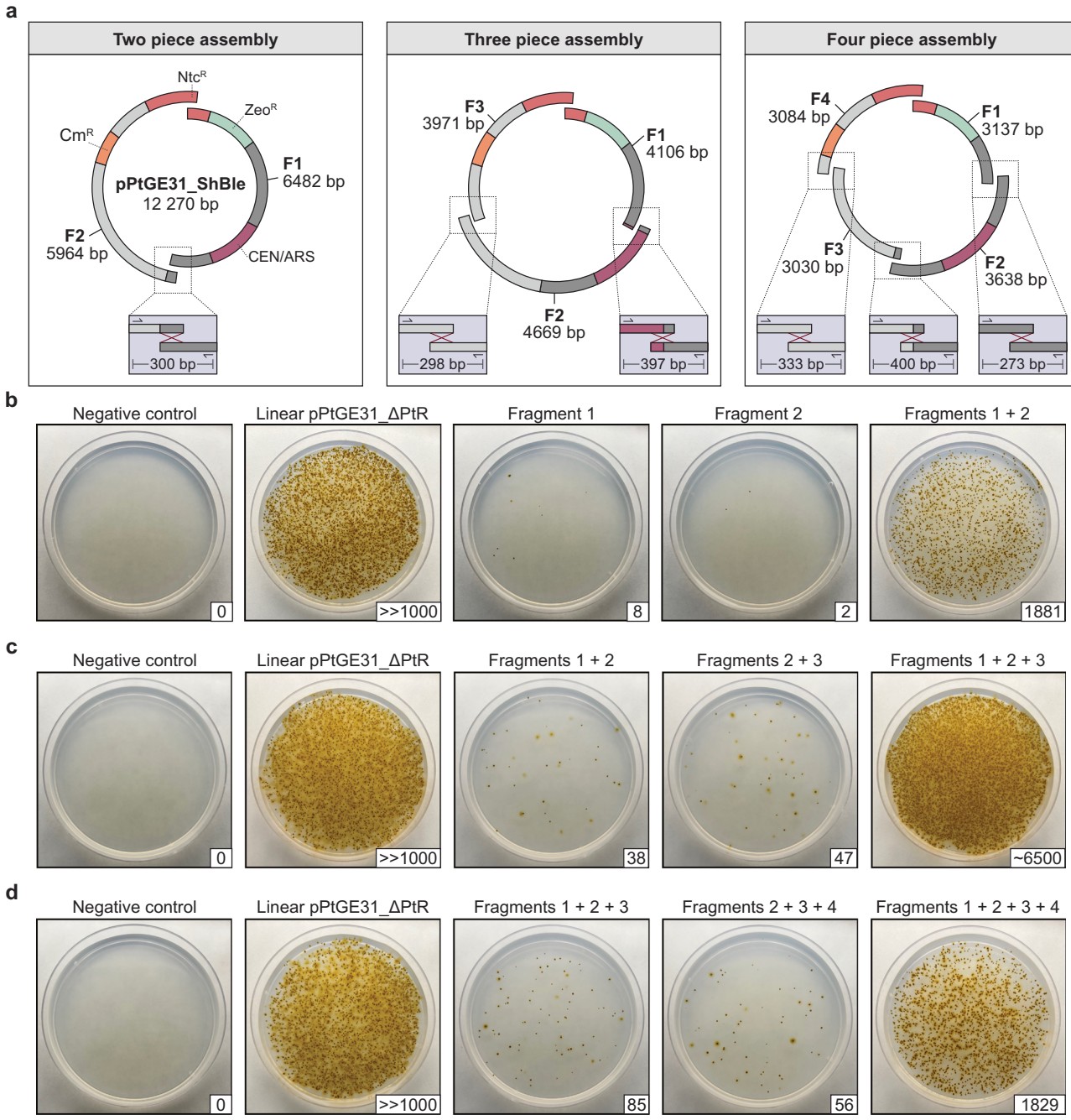

**Fig. 5 | Electroporation and assembly of overlapping fragments in *P. tricornutum*. a** The episome pPtGE31_ShBle was amplified as two, three, or four overlapping fragments (F1: fragment 1, F2: fragment 2, F3: fragment 3, F4: fragment 4). The fragments were adjusted to be equimolar before combining them into their respective assembly mixtures. Primers were designed to screen the other overlapping junction(s), as demonstrated in the bottom boxes. The episome contains a bacterial Cm$^R$ marker, the algal Ntc$^R$/Zeo$^R$ markers, and a yeast CEN/ARS. Electroporation of **b** two, **c** three, or **d** four overlapping fragments. For each condition, partial fragment combinations that cannot reconstitute the Ntc$^R$ marker were also electroporated to assess template episome carry-over. Linear pPtGE31_ΔPtR (500 ng) was included as a positive control. Following electroporation, half of each reaction was plated on ¼-salt L1 medium supplemented with 100 µg/ml nourseothricin. Colony-forming units (CFUs) are indicated in the bottom right of each plate.

transformants contained undesirable assemblies of the non-overlapping fragments. To be able to reliably assemble episomes in the algal nucleus, an alternative strategy is needed. We sought to test whether overlapping fragments could be simultaneously transformed and assembled through the HDR repair pathway. It was hypothesized that the overlapping sequences between fragments would enable site-specific recombination at the termini, thereby creating episomes of an expected size and directionality.

To test this hypothesis, we split the episome pPtGE31_ShBle into two, three, or four fragments that overlapped with each other by ~160 bp (Fig. 5a). This length was chosen somewhat arbitrarily as a starting point, as we were uncertain about the minimum or maximum overlap size needed for successful HDR. Fragments were generated by PCR amplification using PtGE31_ΔPtR (Addgene ID: 236260) or pPtGE31_ShBle (Addgene ID: 236261) as template DNA (primers listed in Supplementary Data 2). In all assembly designs, one of the overlaps was positioned within the open reading frame (ORF) for

nourseothricin N-acetyl transferase (NAT), which confers nourseo-thricin resistance. Thus, when the assembly fragments are electro-porated, nourseothricin-resistant transformants should only appear if homologous recombination has occurred within the NAT ORF or if there is carry-over of the PCR template DNA. Primers were designed to screen the other overlapping termini for successful recombination (Fig. 5a, bottom boxes).

We first tested the two-piece HDR assembly by electroporating fragments 1 and 2 individually or simultaneously (Supplementary Fig. 5b). When electroporated individually, fragments 1 and 2 gave rise to 8 and 2 CFUs, respectively, indicating that carry-over of the PCR template DNA had occurred. However, when the fragments were electroporated simultaneously, there were more than 1800 CFUs – far more than what would be anticipated by carry-over alone. Ten algal colonies were selected for screening of the other fragment junction (Supplementary Fig. 9a), all of which demonstrated the expected 300 bp band, suggesting that successful homologous recombination had occurred. Episomes from algal colonies C1, C2, and C3 were recovered in *E. coli*. Three *E. coli* colonies were screened per initial algal colony, all of which demonstrated the correctly sized band (Supplementary Fig. 9b). To confirm if the fragments had been correctly recombined, episomes from *E. coli* colonies C1-E1, C2-E1, and C3-E1 were sent for whole plasmid sequencing. This demonstrated that colonies C1-E1 and C3-E1 contained the correctly assembled episome (Supplementary Fig. 10), but the episome recovered from C2-E1 had two partial copies of fragment 1.

The same methodological approach was used to electroporate three overlapping fragments simultaneously (Fig. 5c). Electroporation of fragments 1 and 2 or 2 and 3 gave rise to 38 and 47 CFUs respectively, whereas electroporation of all three fragments gave rise to ~6500 CFUs. Ten algal colonies were screened for the junctions between F1/F2 or F2/F3. All colonies contained the expected bands except for C1 and C2, which lacked an amplicon for the F1/F2 junction and demon-strated a weak amplicon for the F2/F3 junction (Supplementary Fig. 9c). We anticipated that assembly may have failed in these colo-nies, but to further investigate this, we attempted to recover the algal episomes from C1 and C2 in *E. coli*, along with the positively screened episomes from C3 and C4. We were able to recover *E. coli* transfor-mants for all the tested episomes and decided to screen two CFUs per initial algal colony, all of which demonstrated correctly sized ampli-cons for both junctions (Supplementary Fig. 9d). Episomes isolated from *E. coli* colonies C1-E1, C2-E1, C3-E1, and C4-E1 were sent for whole plasmid sequencing, with only C1-E1 deviating from the expected assembly (Supplementary Fig. 10).

To further test the limits of the HDR assembly pathway, we attempted electroporation of four overlapping fragments simulta-neously. Electroporation of only three of the four fragments (i.e., F1 + F2 + F3 or F2 + F3 + F4) gave rise to 85 and 56 transformants, demonstrating the carry-over of template DNA once again (Fig. 5d). However, when all four fragments were combined and transformed simultaneously, there was a > 20-fold increase in the number of CFUs. Ten algal colonies were screened across the remaining three junctions, and all but C1 and C7 demonstrated the expected amplicons (Sup-plementary Fig. 9e). Colonies C1 and C7 specifically lacked the 273 bp amplicon for the junction between F1/F2. Algal episomes from C1, C2, C3, and C7 were transformed into *E. coli*; however, C7 did not give rise to any bacterial transformants, suggesting that incomplete assembly had occurred. Two *E. coli* colonies were screened for the episomes derived from C1, C2, and C3, all of which demonstrated the expected amplicons for the three junctions (Supplementary Fig. 9f). Whole plasmid sequencing of C1-E1, C2-E1, and C3-E1 revealed that correct assembly had occurred in algal colonies C1 and C3 (Supplementary Fig. 10), but C2 contained multiple partial copies of fragments 1 and 4.

In total, 26 of the 30 algal colonies screened across the various assemblies demonstrated that recombination had successfully occurred across the tested junctions (Supplementary Fig. 10). From the episomes recovered in *E. coli*, 7 out of 10 had the correctly assembled episomes when fully sequenced. This demonstrates that it is both feasible and reliable to assemble episomes through the HDR pathway in *P. tricornutum*.

### Exploring the limits and applications of HDR assembly

The discovery of DIVA prompted us to further investigate the practical boundaries and potential applications of this assembly method. We first sought to determine the size of overlaps required to direct HDR. Our earlier assemblies relied on ~160 bp overlaps, but in *Sacchar-omyces cerevisiae*, as little as 20 bp is sufficient for recombination[25]. To test whether this was also true in *P. tricornutum*, we PCR-amplified NAT using primers that generated 0, 20, or 40 bp of overlap with the FcpD and FcpA promoter/terminator pair (Supplementary Fig. 11a). We then PCR-amplified pPtGE30 (Addgene ID: 236256) in two fragments, positioning the termini such that the NAT constructs would assemble into the *Sh Ble* locus, which uses the same promoter/terminator elements.

The NAT constructs were electroporated both individually and with the episomal fragments (Supplementary Fig. 11b). Electroporation of NAT-0, NAT-20, and NAT-40 alone confirmed that the marker with up to 40 bp of promoter/terminator sequences does not confer nourseothricin resistance. The NAT-0 assembly produced no trans-formants, whereas both the NAT-20 and NAT-40 assemblies yielded ~100 transformants. These results indicate that overlaps as short as 20 bp are sufficient for HDR-mediated assembly.

We next examined whether de novo synthesized constructs could be inserted directly into episomes in *P. tricornutum*. We commercially synthesized two constructs: (i) CytB-eGFP (833 bp), which carries the CytB mitochondrial targeting peptide identified by TargetP (2.0)[26], and (ii) eGFP-T2A-NAT$_{2-63}$ (1025 bp), in which eGFP is linked to residues 2 to 63 of the NAT protein via the T2A ribosomal skip sequence[27]. The CytB-eGFP construct was designed to insert into the *Sh Ble* locus of pPtGE31_ShBle (Supplementary Fig. S11c), whereas the eGFP-T2A-NAT$_{2-63}$ construct was designed to insert into a PCR-amplified version of pPtGE31_ΔPtR containing an incomplete NAT cassette (residues 47–188; Supplementary Fig. 11d). In both designs, the synthesized constructs included 50 bp of overlaps at each terminus to their respective insertion sites.

For the eGFP-T2A-NAT design, nourseothricin-resistant colonies should only arise if correct assembly restores the complete NAT gene or if there is unintended carry-over of the template episome. This reduces the likelihood of false positive transformants. Both de novo assembly designs yielded at least one hundred colonies (Supplemen-tary Fig. 11e, f).

Ten nourseothricin-resistant colonies from each assembly were passaged and screened for GFP expression by confocal fluorescence microscopy. Of the CytB-eGFP transformants, 4/10 showed moderate mitochondrial eGFP localization, while all 10/10 of the eGFP-T2A-NAT transformants exhibited strong cytosolic eGFP expression (repre-sentative images in Supplementary Fig. 12). Three colonies per design were subsequently recovered in *E. coli* and sequenced (C3, C5, C7 for CytB-eGFP; C1, C2, C3 for eGFP-T2A-NAT). All recovered episomes contained the corresponding de novo construct; however, only one CytB-eGFP episome (C5-E1) and one eGFP-T2A-NAT episome (C1-E1) displayed the expected assembly configuration. The remaining epi-somes showed unexpected recombination events, including combi-nations of multiple episomal fragments and, in some cases, insertion of multiple copies of the de novo construct or unaligned DNA.

### Delivery of RNPs via the optimized electroporation protocol

RNP complexes can be delivered to *P. tricornutum* through biolistic bombardment[28], providing a means for DNA-free genome engineering. We anticipated that it would also be possible to electroporate RNPs

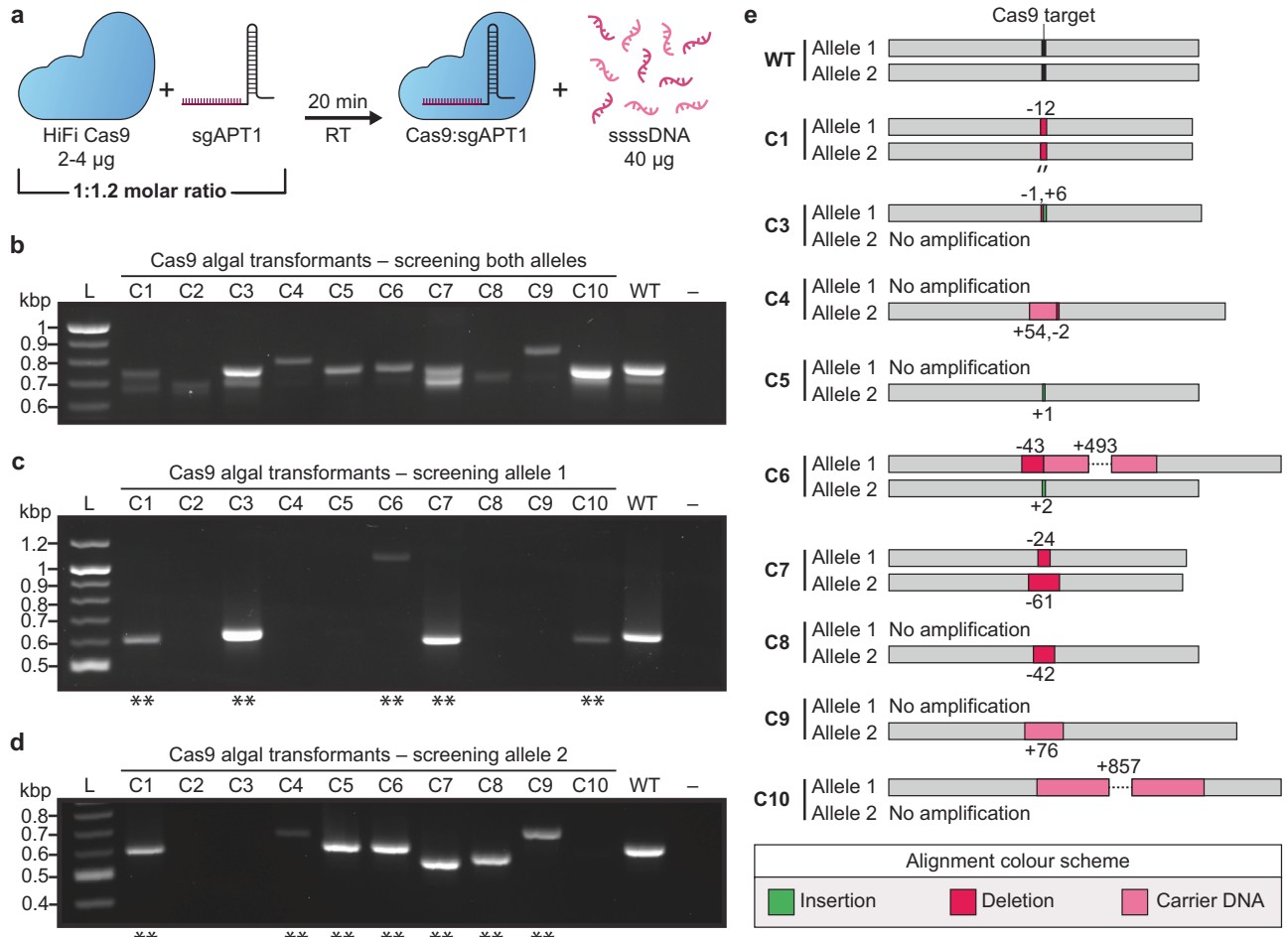

**Fig. 6 | Electroporation of a Cas9:sgAPT1 complex into *P. tricornutum* for DNA-free engineering. a** The HiFi Cas9 complex is combined with an sgRNA that targets the *PtAPT* locus (sgAPT1). After twenty minutes at room temperature, the complex is combined with 40 µg of carrier DNA (ssssDNA) prior to electroporation. **b** Screening ten 2-FA resistant algal transformants with primers that bind to both

*PtAPT* alleles, **c** allele 1, and **d** allele 2. The positive and negative controls consist of wild-type *P. tricornutum* DNA and water, respectively, for all PCR screens. Double asterisks (**) indicate the amplicons that were sent for Sanger sequencing. **e** Alignment of the Sanger sequences captured for algal colonies C1 and C3-C10.

into this species, but this had yet to be reported in any diatoms. As a proof-of-concept, we replicated the methodology Serif et al. had used to deliver RNPs[28] using our optimized electroporation protocol in lieu of biolistics.

Serif et al. had previously shown that knock-out of the *P. tricornutum Adenine Phosphoribosyl Transferase* (*PtAPT*) gene confers resistance to the cytotoxin 2-fluoroadenine (2-FA) at a dose of 10 µM[28]. PtAPT functions in the adenine salvage pathway, catalysing the transfer of phosphoribosyl pyrophosphate (PRPP) to adenine to form adenosine monophosphate (AMP); in the presence of 2-FA, PtAPT instead converts 2-FA into 2-fluoro-AMP, which is subsequently phosphorylated to form toxic fluorinated nucleotides[28].

Chemicals can vary based on the manufacturer and/or batch; thus, when setting forth on this series of experiments, we first tested the resistance of wild-type *P. tricornutum* (CCAP 1055/1) cultured on ½-salt L1 plates supplemented with adenine (5 µg/ml) and our batch of 2-FA (0, 2.5, 5, 10, 15, 20 µM). To our surprise, there was an absence of growth on the 0 µM 2-FA plate, suggesting that adenine alone posed a toxicity to wild-type *P. tricornutum* cells. We tested this theory by plating wild-type cells onto ½-salt L1 plates supplemented with varying levels of adenine (0, 1, 2.5, 5 µg/ml; Figure S13). The growth of wild-type algae was negligible after one-week on the 5 µg/ml adenine plate, but relatively unimpacted on the 0.5 and 1 µg/ml plates. This interesting revelation led us to hypothesize that adenine supplementation could

negatively impact Δ*PtAPT* transformants as well, so when conducting our RNP electroporation experiments, we plated transformants on 2-FA plates with and without adenine supplementation.

The *PtAPT* locus was targeted using two guide RNAs (gRNAs), each of which allowed for biallelic knock-out; we opted to use the first gRNA (gAPT1) for our proof-of-concept experiments as it had generated 11 of the 16 2-FA-resistant transformants in the singleplex experiments by Serif et al.[28]. HiFi Cas9 and a single guide RNA for APT1 (sgAPT1) were complexed according to the manufacturer's guidelines (Fig. 6a). Then, the RNP complex was combined with ssssDNA and electroporated into spheroplasted *P. tricornutum* cells. We opted to include carrier DNA when performing the RNP experiments as ssssDNA is thought to bind to and neutralize the positive charge of the cell membrane during electroporation, making it easier for other charged biomolecules to pass through[29].

Transformants were recovered for one day before plating one-fifth of the reaction across ½ salt L1 plates supplemented with 10 µM or 5 µM 2-FA, with or without 5 µg/ml adenine (Supplementary Fig. 14). This pilot experiment generated thousands of 2-FA-resistant transformants across the plates that had been supplemented with adenine, but only a handful across the plates without supplemental adenine. This was an unexpected discovery given our previous observations. Ten 2-FA-resistant colonies were passaged for further screening at the *PtAPT* loci to determine if Cas9-mediated engineering had occurred.

We used the primer sequences reported by Serif et al.[28] (Supplementary Data 2) to screen the *PtAPT* alleles using PCR. When using primers that bind to both alleles, amplicons of varying sizes were visible across the 2-FA resistant transformants, most of which differed from that of the wild-type strain (Fig. 6b). Likewise, amplicons of varying sizes were visible when using primers specific to allele 1 (Figs. 6c) and 2 (Fig. 6d). This suggested that Cas9-mediated engineering had occurred, but to corroborate our findings, amplicons generated with allele-specific primers were sent for Sanger sequencing.

Alignment of the sequenced amplicons to the wild-type sequence for alleles 1 and 2 confirmed the presence of indels in all 12 alleles that could be analysed (Fig. 6e). We were able to generate sequencing data of both alleles for three of the transformants (C1, C6, and C7), with one clone (C1) demonstrating a homozygous 12 bp deletion across both alleles. Interestingly, there were large (>50 bp) insertions in four of the sequenced alleles. Querying the insertion sequences in NCBI BLAST revealed that carrier DNA had integrated into these loci during the double-strand break repair process. The alignments for C6 and C10 contained two and four non-contiguous regions from the salmon and/or trout genomes, demonstrating that multiple shorn pieces of carrier DNA had been integrated as opposed to a single large fragment.

## Cotransformation of an RNP complex and episome

With the discovery that RNP complexes can be electroporated into *P. tricornutum*, we next sought to see if both an RNP and episome could be co-transformed simultaneously, and if so, what the rate of co-transformation is. The aim of this experiment was to develop a protocol that could significantly improve screening for gene deletions that do not produce an obvious phenotype. To test this, the PCR-amplified episome pPtGE31_ΔPtR was combined with the Cas9:sgAPT1 complex and electroporated into spheroplasted *P. tricornutum* cells. Transformants were first plated on nourseothricin or 2-FA selective plates, and then later repatched onto the alternative selection type to determine if co-transformation had occurred.

Our earlier RNP transformation experiment (Supplementary Fig. 14) had demonstrated that adenine supplementation is important when selecting for *PtAPT* knock-out colonies post electroporation; however, adenine is also seemingly toxic to untransformed *P. tricornutum* cells (Supplementary Fig. 13). Thus, when co-transforming pPtGE31_ΔPtR and Cas9:sgAPT1, the electroporated cells were plated on two different selection plates: (1) ½-salt L1 plates supplemented with 5 µg/ml adenine and 10 µM 2-FA, and (2) ½-salt L1 plates supplemented with 1 µg/ml adenine and 100 µg/ml nourseothricin. The lower adenine concentration was chosen to allow for the growth of co-transformed colonies whilst not unintentionally selecting against colonies that had only uptaken the episome.

After two weeks of growth on nourseothricin or 2-FA selection, 50 co-transformed colonies were repatched onto plates with the same selection type and grown for 1 week. Then, the colonies were repatched onto the alternative selection type to determine if any had double-resistance. Of the 100 screened colonies, 13 demonstrated double-resistance to nourseothricin and 2-FA (Supplementary Table 5), indicating co-transformation had successfully occurred at a rate of ~10%.

## Revisiting the PEG transformation method

A polyethylene glycol (PEG) transformation method for *P. tricornutum* with an efficiency of ≤ 1 transformant per $10^8$ cells was previously described by Karas et al.[6]. For this method, algal cells were protoplasted using a combination of zymolyase, lysozyme, and hemicellulose ahead of PEG treatment. We revisited this method using alcalase as the sole enzyme for cell wall removal, with the intention of providing a simple and equipment-independent alternative to electroporation that is particularly useful for labs or species where this method remains a practical barrier.

We followed the method described by Karas et al.[6], substituting the aforementioned enzyme cocktail with 10, 100, or 1000 µl of alcalase (i.e., 30 mAnson, 300 mAnson, and 3 Anson, respectively) for protoplasting ~3 ×$10^8$ cells. The 3 Anson treatment was potent enough to protoplast the majority (i.e., >80%) of the cells. As well, the protoplasts were transformed with 500 ng of linear pPtGE31_ΔPtR as opposed to 25 µg of a circular episome, the amount originally used by Karas et al., to be able to compare the efficiency of transformation against our electroporation experiments. Treatment with 3 Anson of alcalase generated approximately 160 transformants per 3 ×$10^8$ cells, an efficiency of ~4 transformants per $10^7$ cells. The other two treatments yielded 0 transformants (30 mAnson of alcalase) and approximately 6 transformants (300 mAnson alcalase), so all future PEG transformations were conducted with 3 Anson of alcalase per 3–5 ×$10^8$ cells. This suggests that complete cell wall removal is paramount to efficient PEG transformation.

We then sought to test other parameters of the PEG method to further optimize the transformation efficiency. First, we began to include 40 µg of carrier DNA per PEG reaction as this had led to a ~15-fold increase in transformation efficiency during electroporation. Then, we tested various concentrations of PEG-8000 and found that a range from 20 to 30% (w/v) can be used (Fig. S15a). Going forward with 20% PEG, we then tested if the duration of protoplasting, inclusion of carrier DNA, and length and/or temperature of incubation would alter transformation efficiency (Fig. S15b). As anticipated, the inclusion of carrier DNA increased transformation efficiency >10-fold (Supplementary Fig. 15c). Longer protoplasting and incubation times also appeared to increase transformation efficiency to ~6 transformants per $10^6$ cells (Supplementary Fig. 15c).

The optimized PEG transformation protocol was then used to transform 1 µg of pSC5 into protoplasted *P. tricornutum* cells, generating ~60 transformants (Supplementary Fig. 15d). DNA were isolated from four algal transformants and recovered in EPI300 *E. coli*. Bacterial transformants were then assessed for conjugation in the same manner as before. All recovered episomes demonstrated the ability to conjugate (Supplementary Fig. 15e), verifying that pSC5 had been intactly transformed into *P. tricornutum* through the PEG method.

## PEG transformation to *T. pseudonana*

Electroporation parameters can vary drastically between species and are often finnicky to troubleshoot or optimize. To expand the applications of our alcalase-based transformation methods, we sought to adapt the optimized PEG transformation protocol for *T. pseudonana*, a model diatom species that possesses a more typical silicified frustule. Given the differences in cell wall composition between these species, *T. pseudonana* cells were subjected to varying protoplasting conditions (Supplementary Fig. 16a) before PEG treatment. The circular episome pBIG1 (8.2 kb) conferring nourseothricin resistance and carrying eGFP was used in these experiments. All protoplasting conditions led to the emergence of colonies on the nourseothricin selection plates, with the harsher protoplasting conditions (i.e., ≥ 1.5 Anson alcalase) yielding more growth.

Transformants from the 500 µl alcalase treatment plate (i.e., 1.5 Anson, 40 min) were pooled and subjected to fluorescence microscopy to assess if eGFP was being expressed, as this would indicate successful episome transfer. Both autofluorescence and eGFP signals were visible in the transformed cells (Supplementary Fig. 17), demonstrating that episome transfer had occurred.

To further characterize if protoplasting was truly occurring, *T. pseudonana* cells were incubated with 10 or 1000 µl of alcalase (i.e., 30 mAnson and 3 Anson, respectively) for 40 min (Supplementary Fig. 16b and c, respectively) and then imaged via microscopy. Based on these

microscopy images, cells from the 10 μl treatment (i.e., 30 mAnson) showed a protoplasting rate of ~4% (1/28 cells), whereas cells from the 1000 μl treatment (i.e., 3 Anson) demonstrated a protoplasting rate of ~50% (13/20 cells). This finding underscores that alcalase can be used to degrade the cell wall of diatoms harboring silica frustules.

## Discussion

In this paper, we demonstrate that partial or complete removal of the cell wall greatly increases the efficiency of electroporation and PEG transformation in *P. tricornutum*. These optimized protocols enabled us to push the limits and possibilities of transformation in diatoms. Episomes as large as 55.6 kb could be intactly delivered through either approach, and it was possible to electroporate RNP complexes directly to the algal nucleus for DNA-free genome engineering. Perhaps most interestingly, we also discovered that episomes can be directly assembled in the diatom cell with high efficiency – a feat that has been predominantly limited to *E. coli* and *Saccharomyces cerevisiae*, the workhorses of synthetic biology. These methods are not only more efficient than what has been reported in the past[5–7], but present as faster and simpler alternatives to bacterial conjugation for the reliable transformation of episomes.

The cell wall is the first physical barrier to entry when attempting to transform diatoms. It follows that permeabilizing this wall should increase the ability for biomolecules, like DNA and RNPs, to successfully enter the cell. Although spheroplasting and protoplasting techniques are commonly used for transforming other algal and plant species[30,31], these approaches had yet to be fully investigated in diatoms until this study. We found that spheroplasting *P. tricornutum* cells with alcalase prior to conducting electroporation increased the efficiency by ~76- to 280-fold (Supplementary Table 1), with cells derived from plated cultures demonstrating the highest transformation rates. Protoplasting the cells also enabled efficient transformation through the PEG method (~6 transformants per $10^6$ cells).

Though *P. tricornutum* is unique in both its poorly silicified cell wall[32] and willingness to grow on solid media, we were able to adapt the alcalase-based protoplasting technique for *T. pseudonana*, a diatom species with a silicified cell wall that is typically cultured in liquid media. These results suggest that the spheroplasting and protoplasting techniques can be applied to other diatoms, though it will likely be necessary to optimize the intensity and duration of the enzyme treatment on a species-by-species basis along with other parameters in the respective protocols. This expectation aligns with the trajectory of bacterial conjugation in diatoms, which was originally established using plated *P. tricornutum* cells: although initially assumed to have limited applicability, conjugation has since been successfully adapted to multiple species, including *Cyclotella meneghiniana*, *Nitzschia captiva*, and *Fistulifera solaris*[33–35]. This precedent underscores the realistic potential for broader adaptability of the methods presented here.

The surprising discovery of carrier DNA integrated into the termini of a once linear episome led us to explore the potential for assembly directly in *P. tricornutum*. It was possible to simultaneously electroporate multiple episomal fragments that could recombine through NHR or HDR to form a single episome. The HDR pathway is much more robust for assembly purposes, with ~87% (26/30 colonies) screening positively for the recombined fragment junctions and 70% of recovered episomes having the expected sequence during the episomal assembly experiments (Supplementary Fig. 10). This capacity for in vivo recombination led us to coin the term diatom in vivo assembly (DIVA) to describe episome assembly occurring directly within the algal cell.

To further define the practical limits of DIVA, we assessed the minimal homology required for HDR-mediated assembly (Supplementary Fig. 11a, b). Using NAT constructs with 0, 20, or 40 bp of overlap to an episome, we found that 20 bp is sufficient to direct recombination – consistent with observations in *S. cerevisiae* and

indicating that any fragment of interest can be prepared for HDR assembly by adding ≥20 bp of overlaps via PCR.

We next asked whether DIVA could support the direct insertion of de novo synthesized constructs. Two proof-of-concept assemblies – a CytB-eGFP fusion carrying a predicted mitochondrial targeting peptide, and an eGFP-T2A-NAT construct designed to minimize false positives – each yielded ≥100 transformants. Imaging revealed mitochondrial eGFP localization in 4/10 CytB-eGFP transformants and robust cytosolic expression in all 10/10 eGFP-T2A-NAT transformants screened. Sequencing of recovered episomes confirmed integration of the intended constructs, though several recombinants carried unexpected rearrangements, highlighting the need to further define the fidelity and constraints of DIVA.

Together, these results demonstrate that episomes can be assembled de novo in *P. tricornutum* via DIVA and establish a foundation for further exploration of this process. At present, however, the limits and most appropriate applications of DIVA remain unclear. Key parameters like the upper limit on fragment number, the size range of fragments, and the optimal overlap length, will require systematic investigation before DIVA can be deployed as a reliable or generalizable assembly strategy. We anticipate that the clearest advantages of DIVA may emerge in contexts where rapid or pooled assembly directly in *P. tricornutum* is useful – such as screening promoter libraries or constructing variant pools for phenotype-based enrichment.

The balance between homology-driven and non-homologous repair pathways is also likely to influence the accuracy of DIVA. Angstenberger et al. have demonstrated that knockdown of *ligIV*, an essential component of the non-homologous end joining (NHEJ) pathway, significantly increases the efficiency of homologous recombination when integrating a homology-bearing transgenic cassette into the *P. tricornutum* genome[36]. They also identified other potential targets of the NHEJ pathway (i.e., *Ku70/80*, *XRCC4*, and *BRCT*), none of which have yet to be reported in the literature for *P. tricornutum*. Knockdown of these genes may improve the fidelity of HDR-based DIVA, although the DNA repair mechanisms governing episomal assembly may differ from those involved in chromosomal integration.

Past methods for Cas9-based engineering in *P. tricornutum* included the delivery of RNPs through biolistics[28] or the bacterial conjugation of episomes expressing the Cas9 complex[9]. Our paper adds electroporation and PEG transformation as additional tools for Cas9 genome engineering, with it being possible to deliver episomes through either method or directly electroporate RNPs into *P. tricornutum*. We generated thousands of 2-FA resistant transformants by electroporating an RNP complex targeting *PtAPT*, >200-fold more efficient than what was previously reported with biolistics[28]. It was also possible to cotransform an episome and the *PtAPT* RNP complex at a rate of ~10%. This offers an alternative method for engineering regions of the genome that are not counter-selectable, as a proportion of transformants that have uptaken the episome, which can be selected for, will have also uptaken the RNP and incurred genome editing. The capacity to perform multigene engineering by electroporating multiple RNP complexes was not explored in this study, but we believe this will be possible as this has been previously accomplished via biolistics[28].

Through this series of proof-of-concept experiments, we also uncovered an unexpected paradox: while adenine supplementation (5 μg/ml) is visibly toxic to wild-type *P. tricornutum*, it substantially improves the recovery of Δ*PtAPT* transformants under 2-FA selection. Because *PtAPT* functions in the adenine salvage pathway, it is not immediately clear why supplemental adenine would have any beneficial effect on Δ*PtAPT* strains; if anything, one might expect the opposite. This result suggests that additional adenine salvage, recycling, or purine-metabolic pathways may be active in *P. tricornutum*, and that supplemental adenine may mitigate 2-FA toxicity through these alternative routes. Elucidating the underlying metabolic

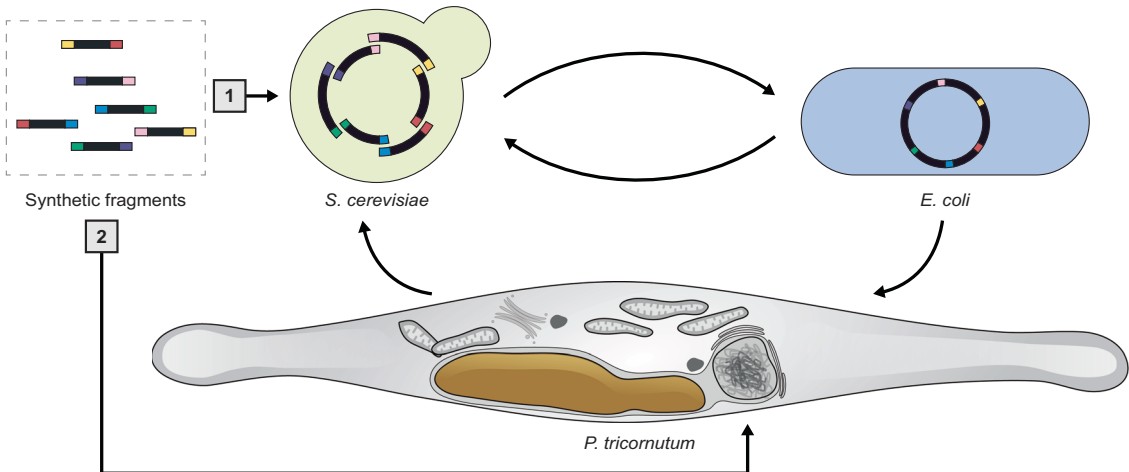

**Fig. 7 | The canonical [1] and alternative [2] pathways for delivering and assembling episomes in *P. tricornutum*.** Prior to this work, overlapping synthetic fragments would be assembled in *S. cerevisiae*. Assembled episomes would be transferred to *E. coli* and then conjugated to *P. tricornutum*. In this pathway, the episome is shuffled between three host organisms during the design-build-test cycle. The alternative pathway provides the opportunity to directly assemble episomes in the agal cell by electroporation of the synthetic fragments.

regulation will require further investigation, but is beyond the scope of the present study.

Complete sequencing data could not be captured for all the analyzed *PtAPT* mutants. This is likely due to the presence of large deletions that extend beyond the range of the primer binding locations. For reference, deletions as large as ~2.7 kb have been observed in Cas9-edited *P. tricornutum* loci before[37] and the primer sets for screening the *PtAPT* alleles were only positioned up to ~450 bp away from the cut site. The alleles that could be sequenced demonstrated that Cas9 engineering had occurred in all assessed clones and revealed the unexpected insertion of carrier DNA in some clones. Based on this finding, we anticipate that it will be possible to intentionally introduce heterologous sequences into the *P. tricornutum* genome by simultaneous electroporation of an RNP along with a selectable DNA fragment bearing homology arms to the targeted locus. Episome-based Cas9 genome engineering has been used to integrate heterologous sequences into the *P. tricornutum* genome before[38], so we anticipate that RNP-based integration of similarly sized heterologous inserts should be possible.

The methods presented in this study pave the way for accelerated engineering in *P. tricornutum*. When we first proposed the Pt-syn 1.0 project, we envisioned using bacterial conjugation to introduce synthetic chromosomes into the algal cell[14]. Now, we can envision the in vivo assembly of synthetic chromosomes directly in the algal nucleus or the piece-by-piece replacement of endogenous chromosomes with synthetic parts (Fig. 7). Challenges and questions remain about the limits of these approaches that future work will have to address before these feats can be accomplished, but in any case, this work has led us one step closer towards actualizing Pt-syn 1.0 – a diatom controlled by a wholly-synthetic genome.

To conclude, we establish improved electroporation and PEG-based approaches that increase the efficiency and reliability of DNA delivery into *P. tricornutum*. These methods allow transformation with very small amounts of DNA, support the uptake of large episomes, and enable both non-homologous and homology-directed assembly of episome fragments directly in the algal nucleus. We also show that Cas9 RNPs can be delivered by electroporation for DNA-free genome editing, and that episomes and RNPs can be co-transformed to facilitate screening of edits that lack obvious phenotypes. Finally, the adaptation of the PEG method to *T. pseudonana* demonstrates the broader applicability of this approach to other diatom species. Together, these results provide an optimized set of tools for diatom genetic

engineering and will support future efforts that require rapid testing and construction of synthetic DNA in algal systems.

## Methods

### Microbial strains and growth conditions

*P. tricornutum* (Culture Collection of Algae and Protozoa, CCAP 1055/1) was cultured in liquid L1 media or on ½-salt L1 1% agar plates (w/v) in a growth chamber set to 18 °C and a light/dark cycle of 16/8 h. Cultures were irradiated with white LED lights (Mars Hydro, model: TSL 2000) set to photon flux densities of ~75 μE m − 2 s − 1 (liquid cultures) or ~55 μE m − 2 s − 1 (agar plate cultures). Agar plates were placed in clear polystyrene trays to reduce desiccation. L1 media was prepared as previously described and lacks silica[6].

In brief, liquid L1 medium was prepared by combining 1 L aquil salts (synthetic seawater), 2 ml NP stock, 1 ml L1 trace metals stock, and 0.5 ml f/2 vitamin solution. Once combined, 2 M NaOH was added until a pH of ~8.0 was achieved and the solution was sterilized through a 0.2-μm filter. For solid medium, sterile liquid L1 medium was mixed 1:1 with autoclaved 2% (w/v) Bacto-agar (all equilibrated to ~60 °C) and poured into petri dishes.

Aquil salts were prepared by mixing equal volumes of separate 2X anhydrous and hydrous salt solutions. The 2X anhydrous salt solution was prepared by dissolving the following in 500 ml ddH$_2$O: NaCl (24.5 g), Na$_2$SO$_4$ (4.09 g), KCl (0.7 g), NaHCO$_3$ (0.2 g), KBr (0.1 g), H$_3$BO$_3$ (0.03 g or 3 ml of a 10 mg ml$^{-1}$ stock), and NaF (0.003 g or 300 μl of a 10 mg ml$^{-1}$ stock). The 2X hydrous salt solution was prepared by dissolving MgCl$_2$·6H$_2$O (11.1 g) and CaCl$_2$·2H$_2$O (1.54 g) in 500 ml ddH$_2$O.

The NP stock solution was prepared at 100X by dissolving NaNO$_3$ (37.5 g per 100 ml) and NaH$_2$PO$_4$·H$_2$O (2.5 g per 100 ml) in ddH$_2$O. The trace metal stock was prepared at 1000X by dissolving the following in 1 L of ddH$_2$O: FeCl$_3$·6H$_2$O (3.15 g) and Na$_2$EDTA·2H$_2$O (4.36 g), followed by addition of the following components from concentrated stocks: CuSO$_4$·5H$_2$O (9.8 g L$^{-1}$), 0.25 ml; Na$_2$MoO$_4$·2H$_2$O (6.3 g L$^{-1}$), 3.0 ml; ZnSO$_4$·7H$_2$O (22.0 g L$^{-1}$), 1.0 ml; CoCl$_2$·6H$_2$O (10.0 g L$^{-1}$), 1.0 ml; MnCl$_2$·4H$_2$O (180.0 g L$^{-1}$), 1.0 ml; H$_2$SeO$_3$ (1.3 g L$^{-1}$), 1.0 ml; NiSO$_4$·6H$_2$O (2.7 g L$^{-1}$), 1.0 ml; Na$_3$VO$_4$ (1.84 g L$^{-1}$), 1.0 ml; and K$_2$CrO$_4$ (1.94 g L$^{-1}$), 1.0 ml. Lastly, the vitamin stock was prepared at 2000X by adding thiamine-HCl (200 mg L$^{-1}$), biotin (10 ml L$^{-1}$ of a 0.1 g L$^{-1}$ stock), and cyanocobalamin (1 ml L$^{-1}$ of a 1 g L$^{-1}$ stock).

*T. pseudonana* (Bigelow National Center for Marine Algae and Microbiota, catalog number: CCMP1335) was cultured in liquid F/2

media in a growth chamber set to 20 °C and 24 hour white light at a photon flux density of 50 µE m$^{-2}$ s$^{-1}$.

*S. cerevisiae* strain VL6-48 (American Type Culture Collection [ATCC], catalog number: MYA-3666) was grown in 2X YPAD media or plated on complete minimal media lacking histidine with 1% agar (w/v) and 1 M D-sorbitol. Yeast cultures were incubated at 30 °C, with liquid cultures placed on a shaker set to 225 rpm. Agar plates were sealed with parafilm to reduce desiccation. Dropout media were prepared according to the manufacturer specifications; in brief, a minimal base media (Takara, catalog number: 630411) was combined with a -HIS dropout supplement (Takara, catalog number: 630415) in ddH$_2$O. Once combined, 2 M NaOH was added until a pH of ~5.6 was reached. YPAD media was sterilized through a 0.2-µm filter; dropout media was sterilized by autoclaving.

*E. coli* strain EPI300 (LGC Biosearch Technologies, Lucigen, catalog number: EC300110) was grown in lysogeny broth (Miller formula) with or without 1% agar (w/v) and supplemented with 15 µg/ml chloramphenicol and/or 40 µg/ml gentamycin and/or 100 µg/ml L-arabinose. Cultures were placed in an incubator set to 37 °C, with liquid cultures placed on a shaker set to 225 rotations per minute (rpm).

## PCR amplification

Episomes and/or fragments for transformation and/or assembly were amplified with GXL polymerase (Takara, catalog number: R050A) using the rapid PCR protocol (all primers are listed in Supplementary Data 2). Of note, we used both phosphorylated and non-phosphorylated primers when amplifying pPtGE31_ΔPtR as a linear episome. PCR reactions were conducted with a total of 25 to 35 cycles depending on amplification efficiency. Fragments that were amplified from an episome template were treated with 10 units (0.5 µl) of *Dpn*I (New England Biolabs, catalog number: R0176) and incubated at 37 °C for 30 min before deactivation at 80 °C for 20 min. After deactivation, fragments were column-purified using the PureLink PCR Purification Kit (Invitrogen, catalog number: K310001) and eluted with ddH$_2$O. Where possible, primers were designed using Primer3 (version 4.1.0).

## PCR screening of algal and *E. coli* transformants

DNA isolated from algal and *E. coli* transformants was screened for the presence of specific episomal regions using a PCR assay. Here, 1 µl of DNA isolated through alkaline lysis was used as template in a 10 µl SuperPlex PCR reaction (Takara, catalog number: 638543). For all algal transformants, episomes were analzyed with a single primer set at a time as there was rampant off-target binding when multiple primer pairs were used simultaneously (all primers are listed in Supplementary Data 2). We were able to successfully conduct multiplex PCR using *E. coli* template DNA; however, when screening the NHR assembled episomes, we chose to perform singleplex screens due to the unexpected recombination of the fragments. Reactions were carried out according to the manufacturer guidelines for a total of 30 cycles.

## DNA isolation

Episomes were isolated from *P. tricornutum*, *S. cerevisiae*, and *E. coli* using an alkaline lysis protocol that was previously described in Karas et al.[6], with species-specific modifications detailed below. Briefly, cell pellets were resuspended in 250 µl of P1 (Qiagen, catalog number: 19051), with or without additional protoplasting enzymes as indicated below, and lysed with 250 µl P2 buffer (Qiagen, catalog number: 19052). Samples were mixed by 5–10 inversions and incubated for 1 min before neutralization with 250 µL P3 buffer (Qiagen, catalog number: 19053). Lysates were centrifuged at 16,000 x G for 10 min, and ~700 µl of supernatant was transferred to a fresh tube. DNA was precipitated from the lysate by adding 750 µl ice-cold 100% isopropanol, which was then centrifuged at 16,000 × g for 10 min. The DNA pellets were washed with 500 µl ice-cold 70% ethanol and centrifuged for

5 min. Pellets were air-dried and resuspended in 30–50 µl sterile ddH$_2$O.

Steps 1 to 2 for *E. coli*: (**1**) *E. coli* colonies were grown overnight in il LB supplemented with the appropriate antibiotic. (**2**) 1.5 to 3.0 ml of culture was pelleted by centrifugation at 10,000 x G.

Steps 1 to 3 for *S. cerevisiae*: (**1**) 3–5 ml of *S. cerevisiae* transformants (pooled) were harvested from dropout media plates ~5 days after assembly. (**2**) The cells were pelleted by centrifugation at 3000 x G for 10 min. (**3**) The P1 resuspension buffer consisted of 240 µl P1 with 5 µl of 1.4 M b-Mercaptoethanol and 5 µl Zymolyase solution (Zymolyase solution: 200 mg Zymolyase 20 T (BioShop, catalog number: ZYM001.100), 9 ml H$_2$O, 1 ml 1 M Tris pH 7.5, 10 ml 50% glycerol, stored at -20 °C). Following resuspension, the cells were incubated at 37 °C for 60 min ahead of the P2 step.

Steps 1 to 3 for *P. tricornutum*: (**1**) *P. tricornutum* transformants were repatched at least twice and grown to high density on a ¼-salt L1 plates supplemented with the appropriate antibiotic(s) (i.e., 100 µg/ml nourseothricin or zeocin for single selection, double selection plates contained both antibiotics at 50 µg/ml). On the second passage, colonies were patched as large streaks to ensure there were enough cells for DNA isolation – typically, a single plate would be used to passage 6 colonies that were struck out to each cover approximately a sixth of the plate. Plates were grown until a reasonable cell density was achieved, which typically took 4 days for nourseothricin plates and upwards of 6 days for zeocin, 2-FA, or double-selection plates. (**2**) Cells were scraped and resuspend in 250 µl of resuspension buffer, which consisted of 245 µl of P1 (Qiagen, catalog number: 19051) and 5 µl of alcalase (i.e., 15 mAnson, Sigma-Aldrich, catalog number: 126741). (**3**) The resuspended cell mixture was vortexed for 1 to 3 seconds, then placed in a heating block set to 56 °C for 10 min – this is within the optimal temperature range for alcalase activity, according to the manufacturer. After 10 min passed, the cells were vortexed again for 1 to 3 seconds to ensure there were no cell clumps.

For *E. coli*-derived episomes smaller than 11 kb, we used column purification kits to isolate and purify DNA (New England Biolabs, catalog number: T1110). DNA was eluted with ddH$_2$O equilibrated to 56 °C.

To isolate a high concentration of pSC5 (~56 kb) from *E. coli*, alkaline lysis was performed using 27 ml of culture (1.5 ml culture per lysis reaction, 18 reactions in total). The DNA pellets were resuspended with ~25 µl of sddH$_2$O and combined into a single 1.5 ml tube ( ~ 460 µl total). Half of the isolated DNA was then digested using *Sac*I-HF, as described below. Then, a tenth volume of sodium acetate (3 M, pH 5.2) and two volumes of ice-cold 100% ethanol were added to the samples. The solutions were inverted to mix and placed into a -80 °C freezer for an hour before performing centrifugation at 16,000 x G for 10 min at 4 °C. The DNA pellets were then washed twice with 500 µl of ice-cold 70% ethanol before being decanted, dried, and resuspended with 50 µl of sddH$_2$O. This additional sodium precipitation step was performed to further concentrate the samples and reduce the carryover of contaminants, particularly from the digest reaction.

DNA concentrations were measured by fluorometry using the DeNovix dsDNA Broad Range Assay (FroggaBio, catalog number: DSDNA-BROAD-EVAL) and purity ratios were assessed using a Nanodrop spectrophotometer. For the experiments shown in Figs. 2 to 4, samples were adjusted to be equimolar prior to conducting electroporation.

## Restriction digests of episomes

Circular pPtGE31_ΔPtR was isolated a *P. tricornutum* transformant and recovered in EPI300 *E. coli*. After isolating the episome from *E. coli*, approximately 10 µg was digested using 3 µl of *Eco*RI-HF (New England Biolabs, catalog number: R3101) at 37 °C for 1 hour. An additional reaction was conducted where 10 µg was digested using 3 µl of *Eco*RI-HF and 3 µl of mung bean nuclease (New England Biolabs, catalog number: M0250) at 37 °C for 1 hour. Complete digestion was

confirmed by agarose gel analysis, then samples were purified using a DNA Cleanup Kit (New England Biolabs, catalog number: T1030).

Circular pSC5 was isolated from EPI300 E. coli as described above. Prior to conducting the sodium acetate precipitation step, approximately 40 μg of pSC5 was digested with 18 μl of SacI-HF (New England Biolabs, R3156) at 37 °C for 1 hour. Complete digestion was confirmed by agarose gel analysis, then the sample was purified by sodium acetate precipitation, as described above.

## Agarose gel electrophoresis

DNA from plasmid isolations or PCR screens were assayed using 1–2% (w/v) agarose gels stained with ethidium bromide. Uncropped images for all gels presented in this manuscript can be viewed in the Source Data file.

## Preparation of Cas9:sgAPT1 complex

HiFi Cas9 nuclease (catalog number: 1081060) and synthetic sgRNA were purchased from Integrated DNA Technologies (IDT) and prepared according to the manufacturer's guidelines. Briefly, 2 to 4 μg of Cas9 (62 μM) was combined with sgRNA (100 μM) in a 1:1.2 molar ratio and then incubated at room temperature for 20 min to form the Cas9:sgRNA complex. After incubation, the complex was set on ice for up to 1 hour, during which the P. tricornutum cells were prepared for electroporation. Once the cells were ready, 40 μg of ssssDNA was added to a 50 μl preparation of the cells, followed by the Cas9:sgRNA complex. The mixture was pipetted up-and-down a few times before transferring to an electrocuvette and promptly pulsed. For the co-transformation experiment, we added 500 ng of PCR-amplified pPtGE31_ ΔPtR to the cells ahead of electroporation.

## Yeast assembly of pPtGE31_ShBle

The episome pPtGE31_ShBle was created by yeast assembly of three overlapping fragments. First, the episome pPtGE31_ ΔPtR was amplified as two fragments using primers that would add 40 bp overlaps for the promoter and terminator region of the ShBle cassette and split the episome in the HIS3 region. Then, the ShBle cassette was amplified from the p0521s plasmid (Addgene ID: 62862)[6] using primers that would add 40 bp overlaps for the insertion site in pPtGE31_ ΔPtR (all primers are listed in Supplementary Data 2). Yeast assembly was conducted as previously described[39] using ≥ 100 ng of each fragment in place of a bacterial donor.

## Electroporation to P. tricornutum

Prior to attempting this methodology, it is worth noting that there is a great degree of variance between electroporation equipment, and this can drastically impact the efficiency of transformation to P. tricornutum. If others attempt this protocol, it may be necessary to initially attempt electroporation with several different parameters to determine which settings work best with different set-ups. For our set-up (500 V, 50 μF, 400 Ω), an empty electrocuvette generates a time constant of ~32 to 35 milliseconds. A neighboring lab has successfully performed electroporation to P. tricornutum using an older electroporator set to the originally described parameters by Zhang and Hu (500 V, 25 μF, 400 Ω)[5].

P. tricornutum cells were grown in liquid media or on agar plates for the experiments conducted in Fig. 1 – all successive experiments used cells derived from agar plates. For liquid cultures, 1 ml of cells adjusted to 1–2 × 10^8 cells/ml were passaged into 50 ml of L1 media and placed in a growth chamber on a shaker set to ~100 rpm. For agar plate cultures, 250 μl of cells adjusted to 1 × 10^8 cells were spread onto a ½-salt L1 plate with 1% agar (w/v). Liquid and agar plate cultures were grown for 4 days. After 4 days, cells were scraped from the agar plates using 750 μl of L1 and transferred to a 1.5 ml tube. An additional 750 μl of L1 was used to remove any remaining cells from the plate. Then, 10^{-1} to 10^{-2} dilutions of the liquid and agar plate cultures were prepared and used to estimate cell density with a hemocytometer. Liquid and agar plate cultures were adjusted to have a final cell density between 2 and 4 × 10^8 cells/ml; for the liquid cultures, this requires concentrating the cells via centrifugation and resuspension in a smaller volume. Here, the cells are transferred from the flask to a 50 ml conical tube and pelleted by centrifugation at 2000 x g for 10 min at 18 °C. The pellet was resuspended with 375 mM D-sorbitol to a concentration between 2 and 4 × 10^8 cells/ml and transferred to a 1.5 ml tube. Cells derived from agar plates were pelleted by centrifugation at 750 x g for 4 min; here, we used a microcentrifuge, which could pellet the cells at a lower centrifugal speed. As before, the cells were resuspended using 375 mM D-sorbitol to a concentration between 2 and 4 × 10^8 cells/ml. The final cell concentration was equalized between the liquid and agar plate-derived cultures for every biological replicate.

To spheroplast, 1 mAnson of alcalase (Sigma-Aldrich, catalog number: 126741) was added for every 1 × 10^8 cells. At this stage, the cells will have been previously concentrated to a density between 2 – 4 × 10^8 cells/ml and already resuspended in 375 mM D-sorbitol. Alcalase was diluted 10-times in sddH_2O to reduce the pipetting error associated with small volumes. After the addition of alcalase, cultures were placed on a rocking shaker set to ~15 rpm at room temperature for 20 min, during which spheroplasting occurs. Once 20 minutes had passed, the cells were pelleted by centrifugation at 325 x G for 4 min. The supernatant is decanted, and the cells are gently resuspended with 1 ml of ice-cold 375 mM D-sorbitol using a P1000 pipette. Following this step, all materials (i.e., cells, DNA, D-sorbitol, electrocuvettes) should be kept on ice for the duration of this protocol.

The electroporation protocol we used is adapted from Zhang and Hu[5], with adjustments made by Kassaw et al.[18] and Pampuch (unpublished). First, the spheroplasted cells were pelleted by centrifugation at 325 x g for 4 min. The supernatant is decanted, and the pellet is gently resuspended with 1 ml of ice-cold 375 mM D-sorbitol. This process of centrifugation and resuspension is repeated another 2 times, such that in total, the cells will have been resuspended with 375 mM D-sorbitol five times (this includes centrifugation steps prior to and post-spheroplasting). After the fifth wash, the cells are spun at 750 x G for 3 min, and a P200 pipette is used to gently remove any remaining supernatant. The cells are gently resuspended with 100 to 200 μl of 375 mM D-sorbitol such that the total volume of resuspended cells will be between ~150 to 250 μl. Into sterile 1.5 ml tubes, 50 μl of cells are aliquoted along with 4 μl (10 μg/μl) of single-stranded salmon sperm DNA (ssssDNA, Sigma-Aldrich, catalog number: D7656) and the DNA to be transferred. Of note, the ssssDNA should be heated to 95 °C for 2 to 5 min and then cooled on ice prior to this to ensure it remains single-stranded. We used between 1 to 1000 ng of an 11 kb episome for most transformations in this paper, with 500 ng of this construct appearing sufficient to saturate the electroporation reaction. Negative controls consisted of only ssssDNA in the cell mixture.

Cells are gently pipetted up and down 2 times before transferring to an ice-cold sterile electrocuvette (Fisher Scientific, catalog number: FB102), ensuring that no air bubbles have been transferred and the cells lie evenly across the bottom. The electrocuvette is pulsed in an electroporator (BioRad, Gene Pulser Xcell System, catalog number: 1652660) set to the following parameters – voltage: 500 V, capacitance: 50 μF, and resistance: 400 Ω. The time constant varied from 30 to 32.5 milliseconds for the experiments conducted in Figs. 1–4. Immediately after pulsing, 1 ml of L1 media is gently added to the electrocuvette without disturbing the cells. The electrocuvettes are left to rest at room temperature for 10 min before fully resuspending the cells and transferring the mixture to 10 ml of L1 in 50 ml Falcon tubes. The lids of the Falcon tubes are left loose for air transfer, and the cultures are moved to a growth chamber to recover for 16 to 24 h.

After recovery, the cultures are pelleted by centrifugation at 2000 x G for 10 min at 18 °C. The pellet is resuspended with 500 μl of L1 media, then 50 μl (one-tenth), 125 μl (a quarter), and/or 250 μl (half) of

the cells are spread onto ¼-salt L1 plates supplemented with the appropriate antibiotic(s).

## PEG transformation of *P. tricornutum*

The optimized PEG transformation protocol is similar to the methods described by Karas et al.[6], with some important modifications. First, 250 μl of wild-type cells adjusted to $1 \times 10^8$ cells were spread onto a ½-salt L1 plate with 1% agar (w/v). After 4 to 5 days of growth, cells were scraped from the agar plates using 1.5 μl of L1 and transferred to a 1.5 ml tube. The cells were counted via a hemocytometer and adjusted to a concentration of $2–4 \times10^8$ cells; in our experience, the density of the scraped cells typically falls within this range.

For protoplasting, 1 ml of cells was transferred to a 50 ml Falcon tube containing 9 ml of L1 and 1000 μl of alcalase (i.e., 3 Anson); here, alcalase is used in place of the zymolyase, hemicellulase, and lysozyme cocktail previously used by Karas et al.[6]. This mixture was then placed on a slow rocking shaker (~15 rpm) and incubated at room temperature for up to an hour. The protoplasting efficiency can be estimated by examining the cells at ≥200x magnification during incubation. The majority of cells (>80%) should have formed protoplasts before proceeding to the next step. In our hands, the protoplasts remain intact during this reaction, likely because the cells are incubated in full-salt L1 medium, which has a high osmolarity. A relatively high volume of alcalase is required compared to the electroporation method, which may reflect reduced enzyme activity in the presence of L1 salts.

After incubation, 40 ml of L1 was gently added to the cells, which were then pelleted by centrifugation at 1500 x *G* for 5 min at 18 °C. The supernatant was carefully decanted, and the cells were gently resuspended with 500 μl of L1 followed by 8 μl of ssssDNA (10 μg/μl). For all further centrifugation steps, the pellet should be resuspended gently to avoid mechanically shearing the protoplasts. Then, 250 μl of protoplasts were combined with episomal DNA (circular or linear, 500 ng to 1 μg) and 1 ml of 20% PEG-8000 (w/v; also contains 10 mM Tris pH 8, 10 mM CaCl$_2$, 2.5 mM MgCl$_2$) equilibrated to 30 °C.

The protoplast:DNA:PEG suspension was gently mixed through inversion and then incubated at RT for 30 min. After incubation, the mixture was centrifuged for 7 min at 1500 x *g*. The supernatant was removed and cells were resuspended with 1 ml of L1, and then transferred into a 50 ml Falcon tube containing an additional 9 ml of L1 (total volume of 10 ml). The cells were then recovered in the growth chamber for ~48 h (i.e., 2 days). After recovery, the cells were centrifuged for 5 min at 1500 x *g* and 18 °C. The supernatant was decanted, and the pellet was resuspended in 500 μl of L1, then 125–250 μl of the cells were plated on ¼ × L1 supplemented with nourseothricin (100 μg/ml). Plates were left in the growth chamber for at least two weeks before being pictured.

## PEG transformation of *T. pseudonana*

Approximately $1 \times 10^5$ cells/ml of *T. pseudonana* was inoculated into f/2 medium. After 4 days of growth (end of exponential phase), cells were pelleted by centrifugation at 1500 x *g* for 10 min and resuspended to a concentration of $3 \times 10^8$ cells/ml. The same transformation protocol was followed as described for *P. tricornutum*; however, the protoplasting step was conducted in f/2 medium with differing amounts of alcalase and incubation lengths (Supplementary Fig. 12a). Approximately 500 ng of the 8.2 kb episome pBIG1 was combined with 40 μg of carrier DNA for transformation. Following PEG transformation, the cells were recovered in 10 ml of f/2 media and recovered for 2 days prior to plating. Post-recovery, the cells were centrifuged for 5 min at 1500 x *g*. The cells were then resuspended in 500 μl of f/2 media, and 250 μl was plated on full-salt f/2 agarose plates supplemented with nourseothricin (100 μg/ml).

## Screening *P. tricornutum* colonies transformed with two fragments – NHR pathway

Non-overlapping fragments from pPtGE31_ShBle were electroporated both individually and simultaneously into *P. tricornutum* (Supplementary Fig. 3). Cells that were electroporated with both fragments were plated onto ¼-salt L1 plates supplemented with 100 μg/ml nourseothricin or 100 μg/ml zeocin. Transformants that appeared following electroporation with both fragments were passaged twice – first, onto media supplemented with the same antibiotic as the initial transformation, and then onto media supplemented with the alternative antibiotic (Supplementary Table 4). Twenty transformants, ten derived from each initial transformation plate, that demonstrated resistance to both nourseothricin and zeocin were passaged for a third time onto double-selection media (¼-salt L1 supplemented with 50 μg/ml nourseothricin and zeocin). DNA isolation was performed using cells from this third passage, followed by PCR screening.

## Screening *P. tricornutum* colonies transformed with multiple fragments – HDR pathway

Overlapping fragments were adjusted to a final concentration of 56 nM, which is four times more concentrated than the linear episome that was used as a positive control in these experiments (14 nM). Different combinations of the fragments were electroporated to measure the amount of carry-over for the template plasmid, pPtGE31_ShBle. For the experiments exploring the limits of HDR assembly, the molarity of the insert was ~4-5x greater than the episomal fragments. The CytB-eGFP and eGFP-T2A-NAT constructs were commercially synthesized from IDT and PCR amplified to generate ample DNA ahead of assembly; the sequences of these constructs are available in the Source Data file, along with the primers in Supplementary Data 2.

In both series of experiments, cells were plated onto ¼-salt L1 plates supplemented with 100 μg/ml nourseothricin and grown for at least two weeks before passaging 10 colonies from the assembly plates onto the same type of selection plate. The repatched colonies were grown for one week before passaging again for DNA isolation and/or microscopy.

## Recovery of *P. tricornutum* episomes in E. coli

Episomes were isolated from *P. tricornutum* transformants as described above. For each electroporation reaction, 2 to 4 μl of *P. tricornutum* DNA was combined with 25 to 50 μl of homemade electrocompetent EPI300 cells (derived from LGC Biosearch Technologies, catalog number: EC02T110) in a sterile 1.5 ml tube. The mixture was gently pipetted twice and then transferred to a sterile 2 mm electrocuvette, ensuring no air bubbles were transferred. The electrocuvette is pulsed in an electroporator set to the following parameters – voltage: 2.5 kV, capacitance: 25 μF, and resistance: 200 Ω. This generated a time constant of 4.9 to 5.2 milliseconds. Electroporated cells were resuspended by pipetting 1000 μl of SOC media up-and-down in the electrocuvette, and then transferred to sterile 1.5 ml tubes to recover at 37 °C and 225 rpm for approximately 1 hour. After recovery, half of the reaction volume (~400 μl) was spread across a 1.5% agar (w/v) LB plate supplemented with 15 μg/ml chloramphenicol (pPtGE31_ ΔPtR and pPtGE31_ShBle recovery) or 40 μg/ml gentamycin (pSC5 recovery), which were then transferred to a 37 °C incubator. Transformants appeared across the plates within 24 h, with there being a range from 4 to ~100 colonies per transformation.

**For recovery of pPtGE31_ ΔPtR and pPtGE31_ShBle episomes.** Single *E. coli* colonies were picked and inoculated into 5 ml of LB media supplemented with 15 μg/ml chloramphenicol. Cultures were grown overnight at 37 °C and 225 rpm. The following day, 500 μl of culture

was inoculated into 4.5 ml of LB media supplemented with 15 µg/ml chloramphenicol and 100 µg/ml L-arabinose. These episomes contain a pCC1BAC backbone, which can be induced to high-copy number replication in EPI300 by adding L-arabinose to the media. Induced cultures were grown for at least five hours prior to episome isolation, with 1.5 ml of culture being used for DNA isolation, as described above.

**For recovery of the conjugative plasmid pSC5.** DNA isolated from eight *P. tricornutum* pSC5 transformants were electroporated into EPI300 cells. Four colonies per transformation were inoculated into 5 ml of LB supplemented with 40 µg/ml gentamycin. Additionally, EPI300 strains carrying pSC5 (positive control) or pSAP (recipient cell line) were inoculated from glycerol stocks into 5 ml of LB supplemented with 40 µg/ml gentamycin (pSC5) or 15 µg/ml chloramphenicol (pSAP). The cultures were placed at 37 °C and 225 rpm. After the cultures reached saturation (-18 h), 125 µl of the pooled *E. coli* transformants and pSC5 cultures were inoculated into 5 ml of LB supplemented with 40 µg/ml gentamycin. For the pSAP culture, 5 ml of culture was diluted into 45 ml of LB supplemented with 15 µg/ml chloramphenicol. The cultures were grown at 37 °C and 225 rpm for 2 h. Then, the cultures were pelleted by centrifugation at 3000 x *g* for 12 min at 10 °C, decanting the supernatant post centrifugation. The pooled transformants and pSC5 cultures were each resuspended with 100 µl of 10% glycerol, whereas the 50 ml pSAP culture was resuspended with 1000 µl of glycerol. Strains were kept on ice during these steps to cease bacterial growth.

For conjugation, 100 µl of the donor cells (transformant pools [x8] and pSC5) were mixed with 100 µl of the recipient cells (pSAP). The cells were mixed by pipetting up and down 4 times before transferring the whole mixture to a 2% agar (w/v) LB plate. Additionally, 100 µl of the recipient was plated by itself to serve as a negative control. Conjugation plates were incubated at 37 °C for 1 hour. Then, plates were scraped using 2 ml of sddH$_2$O. The scraped cells were vortexed for 5 seconds to ensure any clumps were dispersed. Transconjugants were spot plated onto 1.5% agar (w/v) LB plates supplemented with 40 µg/ml gentamycin and 15 µg/ml chloramphenicol with dilutions ranging from $10^0$ to $10^{-8}$.

### Sequencing and alignment
Amplicons generated by amplifying the episome termini region in *P. tricornutum* transformants were sent for Sanger sequencing (London Regional Genomics Centre). Episomes isolated from *E. coli* were sent for Oxford Nanopore whole-plasmid sequencing (Plasmidsaurus Inc. and Flow Genomics). The resulting sequences were manually aligned to the respective template episomes. Regions that could not be manually aligned were queried using a local alignment algorithm (EMBOSS water tool)[40] to search for homology in the episome. Sequences that could not be aligned manually or algorithmically were queried using BLASTn (NCBI)[41].

### Analyses of *P. tricornutum* transformation efficiency
The total number of colonies per transformation reaction was estimated by manually counting the number of CFUs per plate and then multiplying this value by 2 or 10, depending on whether half or a tenth of the total reaction was plated, respectively. In instances where the transformation plate was too dense to reliably count by eye, five 1 cm$^2$ quadrants were sampled from the plate and counted using a dissecting microscope with 10x magnification. Here, the total number of CFUs was estimated by averaging the number of colonies per quadrant and multiplying this value (i.e., colonies per cm$^2$) by the total surface area of the petri dish.

Transformation efficiencies were estimated by dividing the total number of CFUs by the number of cells per reaction. When graphically comparing transformation reactions, all calculations and plotting were done in RStudio (version 2025.05.1 + 513); otherwise,

we used an Excel spreadsheet (version 16.103.4) to keep track of data and tabulate transformation efficiencies (available in the Source Data file).

### Statistics and reproducibility
For experiments in which transformation efficiency was quantitatively assessed (Fig. 1e, f), four to five independent biological replicates were performed to account for variability between electroporation experiments and to enable statistical analysis. For all other experiments, which were designed as qualitative proof-of-concept assays (e.g., assessment of NHR and HDR assembly), at least two independent biological replicates were performed. Where appropriate, representative plate images from a single biological replicate are shown.

Differences in transformation efficiency across the four treatment groups (plate-derived untreated, plate-derived alcalase-treated, liquid-derived untreated, and liquid-derived alcalase-treated) were statistically assessed using a two-sided Kruskal-Wallis test. Pairwise comparisons between selected groups (LU vs. PS, LS vs. PS, PU vs. PS) were performed using Wilcoxon rank-sum tests with Bonferroni correction for multiple comparisons.

### Reporting summary
Further information on research design is available in the Nature Portfolio Reporting Summary linked to this article.

## Data availability
Sequencing information is available in the Supplementary Datasets 3–5. The raw *P. tricornutum* colony counts used to estimate transformation efficiency for the graphs in Fig. 1e, f are available in the Source Data File. All constructs used in *P. tricornutum* are also deposited to Addgene, as referenced throughout the text. Source data are provided with this paper.

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

## Acknowledgements

This work was supported by Natural Sciences and Engineering Research Council of Canada (RGPIN-2018-06172 and RGPIN-2025-05428) awarded to BJK., and funding from the Natural Environment Research Council (NERC) grant NE/Z504130/1 awarded to TM. We would like to acknowledge Drs Graham Peers and Tessema Kassaw for sharing their knowledge on electroporation to *P. tricornutum* with us. Our explorations into this method began with a protocol that was shared by Dr. Kassaw, who assisted us with initial troubleshooting. We would also like to acknowledge Karen Nygard, a microscopy specialist at Western University's Biotron Facilities, who assisted with capturing the microscopic images shown in Fig. 1a, b.

## Author contributions

M.P. and G.T. established the parameters necessary for conducting electroporation in the Karas lab and performed experiments 1 through 79 in Supplementary Data 1. E.J.W. established the spheroplasting technique for electroporation and the protoplasting technique for PEG transformation in *P. tricornutum*. EJW also performed all other *P. tricornutum* experiments detailed in the manuscript. L.D. and Y.L. performed the PEG transformation experiments in *T. pseudonana* and generated the images used for Supplementary Figs. 16 and 17. E.J.W. wrote the manuscript and created all other figures in Adobe Illustrator; M.P., L.D., Y.L., T.M., and B.J.K. edited the manuscript and provided crucial insight. B.J.K. supervised all experimentation done by E.J.W., M.P., and G.T., and assisted with experimental design, data analysis, troubleshooting, and more. T.M. supervised all experimentation done by L.D. and Y.L.

## Competing interests

The authors declare no competing interests.
