## [Transparent Peer Review file · Nature Communications]

Breaking the cell wall for efficient DNA delivery to diatoms

Corresponding Author: Professor Bogumil Karas

A version of this paper was originally rejected for publication by Nature Communications, however that decision was reconsidered after appeal by the authors.

Version 1:

Reviewer comments:

Reviewer #1

(Remarks to the Author)

This manuscript presents a significant advancement in genetic engineering tools for diatoms, focusing on optimized protocols for DNA and ribonucleoprotein (RNP) complex delivery to *Phaeodactylum tricornutum*. The authors demonstrate that spheroplasting via alcalase treatment greatly enhances transformation efficiency for electroporation and PEG-mediated delivery. The study introduces the concept of Diatom In Vivo Assembly (DIVA), showing that episomes, including those up to 55.6 kb, can be assembled directly within diatom cells through non-homologous end joining (NHEJ) or homologous recombination (HR). Additionally, they present an electroporation-based protocol for DNA-free genome editing using Cas9 RNPs, achieving successful gene knockouts and even co-delivery of RNPs with episomes. These developments are positioned to support synthetic biology efforts, such as the Pt-syn 1.0 project, and can be applied to other diatom species, including *Thalassiosira pseudonana*. The work is clearly described, the data are compelling, and the implications for synthetic biology are profound. The following points can be considered to enhance the clarity of the paper:

Page 3, Lines 103–104:

Data confirming the absence of genomic integration and demonstrating that the episome is maintained in a circular form should be provided to support these sentences. “Data now shown” should be avoided.

Page 4, Line 116–:

In order to substantiate the hypothesis that diatom cell morphology and cell wall integrity affect transformation efficiency via electroporation, experiments involving alcalase treatment should be conducted on oval, fusiform, and triradiate cells. While the following section presents data from liquid cultures, which are likely to contain predominantly fusiform cells, this should be discussed in the former section for consistency and clarity.

Furthermore, more detailed microscopic images showing the morphological changes induced by alcalase treatment would improve understanding of this process. What morphological changes were observed in fusiform and triradiate cells upon treatment? Were those the reasons for lower transformation efficiency?

Figure 2e:

Please comment on the absence of a band at around 486 bp in lanes 3, 4, and 5 in the text. Could the integration of the introduced DNA fragments into the genome result in resistance to the selection antibiotics?

Page 16, Line 501–:

To ensure the purpose of this experiment is immediately clear to the readers, it would be helpful to briefly state the purpose of the co-delivery of RNPs and episomes in this paragraph.

Figure 7:

Although Figure 7 suggests the potential for the targeted insertion of foreign DNA at arbitrary loci in the diatom genome, this possibility has not been demonstrated experimentally in the current study. Although this is a plausible and exciting prospect, the figure should be omitted to avoid misleading readers.

Reviewer #2

(Remarks to the Author)

General Comments:

This manuscript presents promising improvements in DNA delivery protocols for *P. tricornutum*, particularly through the use of spheroplasting in combination with electroporation and polyethylene glycol (PEG)-mediated transformation. The authors demonstrate improved transformation efficiency using both circular and linearized plasmids, and successfully apply ribonucleoprotein (RNP) complexes of Cas9 and gRNA for gene editing. Notably, they provide the first, exciting evidence of in vivo DNA fragment assembly in diatoms.

However, in its current form, the manuscript primarily presents an optimization of existing transformation protocols. While the authors demonstrate successful delivery of various constructs into diatom cells, they do not investigate the fate of the introduced DNA within the algal cells—specifically, whether integration into the genome occurs, the copy number and stability of episomal plasmids, or the potential for epigenetic silencing. Additionally, the novel and potentially impactful observation of in vivo DNA assembly in algae is acknowledged but not explored in depth.

As such, the manuscript may be more appropriate for a technically oriented journal. Alternatively, it could be significantly strengthened by expanding on the mechanisms and biological implications of in vivo DNA assembly in diatoms.

Below are specific comments and suggestions for revision:

Major comments:

Overstatements

The authors claim that DNA delivery is the major challenge in diatom transformation. While this is true for several diatom species, it does not hold for *Phaeodactylum tricornutum*, which can be efficiently transformed using multiple established methods. In contrast, a more significant challenge in *P. tricornutum* is the frequent epigenetic silencing of transgenes. Furthermore, *P. tricornutum* lacks a silica cell wall, making it particularly well-suited for spheroplasting-based approaches. The inclusion of *Thalassiosira pseudonana* is also unconvincing. As the other main model diatom species, *T. pseudonana* has a well-documented history of successful transformations, and it is unclear how the current protocol represents a major advancement for other species.

The authors further propose that de novo diatom in vivo assembly (DIVA) could replace traditional cloning approaches in *E. coli* and *S. cerevisiae*, which they describe as time-consuming. However, this claim is not well substantiated. The data presented show that in vivo assembly in diatoms is prone to errors, in contrast to the reliable transformation with pre-assembled plasmids. Moreover, assembled episomes still require purification and transformation into *E. coli* for verification. This statement also does not account for the lengthy selection process in diatoms, which typically takes around three weeks, compared to the single-day turnaround in *E. coli*. Instead of positioning DIVA as a general replacement for bacterial or yeast cloning, we suggest that the authors propose a more practical and specific application where DIVA could offer a clear advantage.

Co-transformation with RNP Cas9-gRNA complexes and episomal resistance marker: The manuscript presents this strategy as a solution for targeting genomic loci that lack a selectable phenotype. However, with a reported co-transformation efficiency of only 10%, the majority of resulting colonies (~90%) are false positives. The authors might consider exploring an alternative strategy—such as co-delivery of multiple RNP complexes targeting different loci—similar to the approach described in Serif et al., 2018.

Examples of overstatements:

Line 28: The sentence overgeneralizes the adaptability of the protocol. *Thalassiosira pseudonana* is also a model diatom species with a small genome, rapid growth, and generally accessible for genetic transformation. Given the differences between species and that electroporation was only tested in *P. tricornutum* (and most effective on plates), caution should be used when implying broad adaptability.

Line 64: “Conjugation requires the creation and maintenance of a donor strain of *Escherichia coli*, which must contain a suitable broad-host-range conjugative plasmid in addition to the episome of interest.” While this statement is technically accurate, it is unclear why the authors present it as a drawback. The creation and maintenance of donor *E. coli* strains is a standard molecular biology procedure and typically not considered a major limitation.

Line 89: Reiteration of line 28 comment: the claim that these methods can be broadly applied to other diatoms should be more cautious and supported with data or omitted.

General note: The data suggest that spheroplasting improves transformation mainly on plates, with limited benefit in liquid culture. This is important because many diatom species cannot grow on solid media.

Uneven quality across the manuscript:

- The final two sections—PEG transformation and *Thalassiosira pseudonana* transformation—are notably underdeveloped compared to the rest of the experimental work. These sections appear rushed, lacking sufficient detail, proper quantification, and clear presentation of results. To maintain consistency and clarity, the authors should either expand and revise these sections to match the depth and quality of the earlier parts of the manuscript or consider omitting them altogether.

Open questions:

Would the addition of telomeric sequences help stabilize the linearized plasmids and prevent their circularization in diatom cells?

Could the authors clarify why adenine supplementation is necessary following PtAPT knockout? What is the underlying metabolic or genetic rationale?

Additional line-by-line comments:

Line 26: The term “PEG” is used in the Abstract without being defined. Please define “polyethylene glycol” at first mention.

Line 63: The transformation rate is incorrectly written as “4 transformants per 10^{-4} cells”. It should read “4 transformants per 10^4 cells” for consistency with line 58.

Line 69: More background and context on PEG-mediated transformation in diatoms is needed. Currently, the discussion is too brief for readers unfamiliar with the technique.

Line 70: Correct “ ≤ 1 transformant per 10^{-8} cells” to “ ≤ 1 transformant per 10^8 cells”.

Line 87: The manuscript inconsistently uses the terms spheroplasting and protoplasting. These are sometimes used interchangeably and sometimes as distinct processes. Given the title refers specifically to “spheroblashed cells,” a clear and consistent definition is essential.

Line 97: The authors note that a neighbouring lab used older electroporation parameters and that the switch from 25 to 50 μF may relate to the equipment used. This raises the question of whether the improved transformation efficiency is protocol-specific or simply equipment-dependent. Please clarify.

Line 101: Plasmid pPtGE31_ΔPtR with ID 236260 is not found on Addgene. Please upload the plasmid map and sequence before any resubmission.

Line 176: The use of 500 ng of a PCR-amplified 11-kb construct should be mentioned earlier in the text, especially when referring to Figure 1.

Line 256: The claim that linear pSC5 is three times more efficient than circular is not clearly supported by Figure 3B, where colony counts appear similar (5 vs. 6 CFUs). Please clarify.

Line 302: Plasmid pPtGE31_ShBle with ID 236261 is not found on Addgene. Please upload the plasmid map and sequence before any resubmission.

Line 393: There is no Figure S6C. This likely refers to Figure S7C—please correct.

Line 539: A brief conclusion summarizing the main findings and potential applications would strengthen the end of the manuscript.

Line 550: The difference in DNA quantities used (500 ng linear vs. 25 μg circular) is substantial. Please explain how these values were chosen.

Line 554: Early in the manuscript, alcalase dosage is reported in mAnson units per 10^8 cells, but here it is given in μl . Please standardize units throughout.

Line 558: The text states that 40 μg of carrier DNA improved electroporation efficiency ~15-fold, but no results are shown for PEG transformation. Was this amount used in PEG protocols? Was a similar improvement observed?

Line 565: The final optimized PEG protocol is not clearly presented. Please summarize the final conditions used and their effectiveness.

Line 566: There is no Figure S2D, despite the reference. Please correct or include the figure.

Lines 565–571: This paragraph lacks quantitative data and clarity. It is difficult to assess the advantage of PEG transformation without specific efficiency data.

Line 574: Reiterate the need to clearly outline the final optimized conditions.

Line 577: Please include the size of the episome used here, as done in the Materials and Methods section.

Lines 578–581: The results described are vague. Clarify what “some degree of growth” means and how harsher protoplasting conditions influenced colony formation.

Line 588: The phrase “a higher proportion of protoplasted cells are visible...” is vague. Provide more concrete observations or quantification, if available.

Line 593: There is no discussion on partial vs. complete cell wall removal. Is this dependent on enzyme concentration? Do cells survive full wall removal?

Line 611: It is true that plated cells yielded the highest transformation rates, but based on Table S2, efficiencies are mostly in the 10^6 – 10^7 range, with only treated plated cells showing higher values (1.19×10^4). The benefit of alcalase treatment seems to apply mainly to plated cells.

Line 612: The manuscript states that transformation is “efficient,” but no numerical threshold is provided. Please define what constitutes efficiency in this context.

Line 645: “Several-fold more efficient” should be quantified.

Line 701: Only PEG transformation was adapted to *T. pseudonana*. The claim that these methods hold great promise for other diatoms lacks sufficient supporting evidence.

Line 866: The note about keeping materials on ice is useful and should be moved earlier in the protocol section for clarity.

Line 880: The DNA input range (“1 to 1000 ng”) is very broad. Consider adding a note that for ~11-kb episomes, 500 ng appears to saturate the reaction.

Line 904: This line again references equipment differences affecting electroporation settings. Consider addressing this earlier in the Methods for better context.

Figure Comments:

Figure 1b: Why is the image shown with 100 mAnson if the best treatment was achieved with 1 mAnson? This is confusing and needs explanation.

Figure 1e and f: Statistical analysis should be included. Please clarify what alcalase treatment was used and provide full experimental conditions (e.g., DNA concentration, treatment duration) so that the figure is self-contained.

Figure 1e: As noted above, experimental details (e.g., DNA concentration, alcalase protocol) are missing. These are essential for interpretation.

Figure 1f: Clarify whether this experiment was done in liquid culture or on plates, and specify the alcalase treatment used.

Figure 2c–e: Please clarify the annotations in panels d and e to clearly indicate which PCR primer pairs were used, with explicit reference to panel c.

Figure 3e is not clearly explained. Presumably, it should refer to line 269: “conjugation was successful for seven out of eight *E. coli* pools”. If it is the case, then labelling C1–C4 in both cases is confusing.

Figure 5: Part a shows three schemes of possible assembly. Part b does not correspond to part a. Please adjust for clarity.

Reviewer #3

(Remarks to the Author)

I co-reviewed this manuscript with one of the reviewers who provided the listed reports. This is part of the Nature

Communications initiative to facilitate training in peer review and to provide appropriate recognition for Early Career Researchers who co-review manuscripts.

Version 2:

Reviewer comments:

Reviewer #1

(Remarks to the Author)

The authors have addressed my peer review comments appropriately. I have no further points to raise.

Reviewer #2

(Remarks to the Author)

We would like to thank the authors for carefully addressing our previous comments. The manuscript is now substantially improved, and we particularly appreciate the more in-depth analysis of in vivo assembly in diatoms, which opens exciting new avenues for future research. While we have one major comment and a few minor remarks, we consider this work a valuable resource for the diatom research community and believe it deserves publication in Nature Communications.

Main comment

While the work on *Phaeodactylum tricornutum* is very convincing, the transformation of *Thalassiosira pseudonana* still raises some questions.

The circular episome pBIG1 (8.2 kb) conferring nourseothricin resistance and carrying eGFP was used in these experiments.

- Is the sequence of this plasmid available or will it be made available?
- Which promoter drives the expression?
- Does it express only eGFP, or are there other expression elements present?

Although in most cases the alkalase units were standardized to mAnson, in this part of the manuscript, alkalase is still in μ l alkalase

In Supplementary Fig. S17, the statement “Both autofluorescence and eGFP signals were visible in the pooled transformed cells, demonstrating that episome transfer had occurred” is difficult to evaluate. The figure is challenging to interpret: although there is no GFP signal in the untransformed control, the green signal observed in the transformed sample appears in structures lacking chloroplast autofluorescence and not corresponding to *T. pseudonana* cells. This raises concerns about what is being visualized and whether the image may be overexposed.

Minor remarks

- Although the authors have corrected most instances of “ μ l” to proper enzyme activity units for alkalase, a few remain, mainly in the *T. pseudonana* section.
- Line 80: “Electroporation has a higher transformation efficiency (2.6 to 4.5 transformants per 10^5 cells^{5,7})” → Please standardize the citation format with line 62, either inside or outside the parentheses.
- Line 354: The correct reference is Supplementary Table S6, not S5.
- Line 635: “develop a protocol” → missing the article “a”.
- Keep the rudimentary Figure 5 with the plates in the main paper, and move the figure with the gels to the Supplementary Information.

Side remark (for future consideration):

It is possible that the 5× telomeric repeat used is not sufficient to establish a fully protective protein complex equivalent to the human shelterin complex. For future experiments, the authors might consider testing longer telomeric repeats, although these can be difficult to clone and maintain in bacteria.

Reviewer #3

(Remarks to the Author)

RESPONSE TO REVIEWERS

Our responses are in purple.

In addition to the comments provided by Reviewers 1 to 3, we also had a colleague, Dr. David Edgell, read our manuscript. We have made additional changes in accordance with Dr. Edgell's suggestions; notably, it was not correct to refer to the assembly methods as non-homologous end joining and homologous recombination. A better term for these processes is non-homologous repair and homology-directed repair, since we do not know the exact underlying DNA repair process governing episomal assembly in *P. tricornutum*. We also caught some additional small errors upon carefully assessing the manuscript.

Reviewer #1 (Remarks to the Author):

This manuscript presents a significant advancement in genetic engineering tools for diatoms, focusing on optimized protocols for DNA and ribonucleoprotein (RNP) complex delivery to *Phaeodactylum tricornutum*. The authors demonstrate that spheroplasting via alkalase treatment greatly enhances transformation efficiency for electroporation and PEG-mediated delivery. The study introduces the concept of Diatom In Vivo Assembly (DIVA), showing that episomes, including those up to 55.6 kb, can be assembled directly within diatom cells through non-homologous end joining (NHEJ) or homologous recombination (HR). Additionally, they present an electroporation-based protocol for DNA-free genome editing using Cas9 RNPs, achieving successful gene knockouts and even co-delivery of RNPs with episomes. These developments are positioned to support synthetic biology efforts, such as the Pt-syn 1.0 project, and can be applied to other diatom species, including *Thalassiosira pseudonana*. The work is clearly described, the data are compelling, and the implications for synthetic biology are profound. The following points can be considered to enhance the clarity of the paper:

We sincerely thank Reviewer 1 for their positive and encouraging feedback. We are grateful that they found the manuscript to be of interest and appreciate their constructive feedback for enhancing clarity. We have carefully considered each point raised and have revised the text accordingly. Below, we provide a detailed response to each comment.

Page 3, Lines 103–104:

Data confirming the absence of genomic integration and demonstrating that the episome is maintained in a circular form should be provided to support these sentences. “Data now shown” should be avoided.

These are valid points; we should have provided more data to support these statements. To address the first point (i.e., genomic integration), we have performed a plasmid/episome loss experiment by serially passaging four pPtGE31_ΔPtR transformants in non-selective media. The transformants chosen correspond to algal colonies 1 and 2 in Fig. 2D and 1 and 8 in Fig. 2E. After seven passages in liquid non-selective media, serial dilutions of the cultures were then plated onto solid non-selective media to obtain single colonies. For each transformant, 100 single colonies were then re-patched onto plates supplemented with nourseothricin. If genomic integration of the episome had occurred, we would expect a high proportion of the single colonies to retain resistance. Conversely, if the episomes are maintained extrachromosomally, they should be lost over successive generations without selective pressure, and thus only a low proportion of colonies would retain resistance. As expected, only a small fraction of

screened colonies retained resistance (Supplementary Table S4), demonstrating that genomic integration had not occurred in the assayed transformants.

With this in mind, we understand that genomic integration can still occur when transforming episomes, regardless of the delivery method (e.g., conjugation or electroporation); however, this outcome is probabilistically less likely. For integration to successfully yield a nourseothricin-resistant transformant, the construct must not only integrate into a DSB in the genome, but the region of integration must also be transcriptionally active. Comparatively, the episome just needs to make it to the nucleus to yield a successful transformant (generally speaking). This is illustrated in Supplementary Fig. S1, where we transformed equimolar amounts of the episome pPtGE31_ΔPtR (~11 kb) and the nourseothricin resistance cassette (~1.4 kb) into *P. tricornutum* cells. Our work with pSC5 (56 kb) pPtGE31_ΔPtR demonstrated how construct size can significantly impact the efficiency of electroporation. Despite the NAT cassette being nearly tenfold smaller than pPtGE31_ΔPtR, it produced over tenfold fewer colonies, highlighting the lower likelihood of successful genomic integration compared to episomal delivery.

We have clarified this point in the revised manuscript by adding the following section:

“Evaluation of genomic integration

We have performed a plasmid/episome loss experiment by serially passaging four pPtGE31_ΔPtR transformants in non-selective media to check for genomic integration. The transformants chosen correspond to algal colonies 1 and 2 in Fig. 2D and 1 and 8 in Fig. 2E. After seven passages in liquid non-selective media, serial dilutions of the cultures were then plated onto solid non-selective media to obtain single colonies. For each transformant, 100 single colonies were then re-patched onto plates supplemented with nourseothricin. If genomic integration of the episome had occurred, we would expect a high proportion of the single colonies to retain resistance. Conversely, if the episomes are maintained extra chromosomally, they should be lost over successive generations without selective pressure, and thus only a low proportion of colonies would retain resistance. As expected, only 2-7% of screened colonies retained resistance (Supplemental Table S4), demonstrating that genomic integration had not occurred in most assayed transformants.”

To address the second point (i.e., plasmid recircularization), we have added additional data (Supplementary Fig. S2) for clarity. The EPI300 *E. coli* strain cannot propagate non-circular DNA. Therefore, if the algal episomes were maintained linearly, they could not be recovered in EPI300. All of our episomal analyses (e.g., sequencing, conjugation assays) involved recovery of the DNA in EPI300, but to further illustrate this point, we electroporated PCR-amplified pPtGE31_ΔPtR that had been DpnI treated (to get rid of any remnant circular template) into EPI300. This yielded zero colonies, as expected, whereas electroporation of pPtGE31_ΔPtR recovered from algal transformants (which have low copy/poor DNA yield comparatively) yields hundreds of colonies.

In addition, we modified text in result section to address this to read as follows:

“Our initial attempts at electroporation using a previously described protocol¹⁶ that was adapted from the methods of Zhang and Hu⁵ yielded no transformants. After several failed attempts, we discovered that adjusting the electroporator capacitance from 25 to 50 μF yielded transformants for both episomes and marker cassettes, though efficiency was highly variable between experiments (Supplementary Table S1, Exp. 1 to 81). We used this method to deliver a linear 11 kb PCR-amplified episome, pPtGE31_ΔPtR (Addgene

ID: 236260), to assess whether the construct would integrate into the genome, persist as a linear episome, or be converted into a circular episome within the *P. tricornutum* nucleus. We were able to recover DNA from these algal transformants in EPI300 *E. coli*, indicating that the linear episome had been circularized in the algal nucleus following electroporation (Supplementary Fig. S2). Whole-plasmid sequencing of four *E. coli* colonies confirmed that circularization had occurred via fusion of the construct's termini."

Page 4, Line 116–:

In order to substantiate the hypothesis that diatom cell morphology and cell wall integrity affect transformation efficiency via electroporation, experiments involving alcalase treatment should be conducted on oval, fusiform, and triradiate cells. While the following section presents data from liquid cultures, which are likely to contain predominantly fusiform cells, this should be discussed in the former section for consistency and clarity.

We thank the reviewer for this insightful suggestion. Unfortunately, we cannot readily control cell morphology in *P. tricornutum*, and triradiate cells are particularly rare in our strain, making systematic experiments with all morphotypes impractical. We have previously attempted to propagate cultures from single colonies with a particular morphology before, but found that single colonies will differentiate/switch morphotypes rapidly. Nevertheless, our microscopy observations confirm that all cell types can be fully protoplasted with alcalase (can see remnant frustules that appear following the alcalase treatment). We have included additional microscopy images to further illustrate these observations (in accordance with your suggestion below; Supplementary Fig. S3).

We have revised text in the results section to clarify this point to read as follows:

*"When this electroporation protocol was repeated using cells that had reached the late stationary phase of growth, we obtained up to 272 transformants per 10⁸ cells with the same 11 kb PCR-amplified episome (Supplementary Table S1, Exp. 82 to 89). This represented a nearly 20-fold increase in efficiency when compared to transformations with early-log phase cultures (Exp. 80 and 81). Stationary phase cultures demonstrated noticeable differences in cell phenotype when compared to early-log phase cultures, with there being a higher abundance protoplasts (i.e., diatoms lacking a cell wall) in the older cultures. *P. tricornutum* cells can appear in three different morphotypes – fusiform, oval, and triradiate – and can switch morphotypes in response to environmental stimuli^{17,18}. All morphotypes can generate protoplasts if the cell wall is sufficiently weakened. Ultimately, this observation led us to hypothesize that cell morphotype and cell wall integrity may influence electroporation transformation efficiency.*

*Cell morphotype cannot be readily controlled in *P. tricornutum* cultures as this species displays an immense amount of phenotypic plasticity; however, agar-plated cultures of CCAP 1055/1 tend to have a higher proportion of oval-type cells (~20-30% oval, ~70-80% fusiform) compared to liquid cultures (<10% oval, >90% fusiform). Thus, to test this hypothesis, we spheroplasted early-log phase *P. tricornutum* cells harvested from agar plates. Plate-derived cells were spheroplasted using the enzyme alcalase, a serine protease that has been shown to degrade the proteinaceous components of the *P. tricornutum* cell wall¹⁹. We observed that 100 μ l of alcalase at an activity of approximately 3 Anson units per ml (i.e., 300 mAnson units) was enough to fully protoplast 3 x 10⁸ plate-derived cells when resuspended in 375 mM D-sorbitol (Fig. 1A and B). The protoplasted cells were very fragile and prone to rupture from osmotic pressure or physical damage; treatment of 3 x 10⁸ cells with 10 μ l of alcalase (30 mAnson units) was enough to*

protoplast some cells, whilst leaving others with partially degraded cell walls (i.e., spheroplasts)."

Furthermore, more detailed microscopic images showing the morphological changes induced by alcalase treatment would improve understanding of this process. What morphological changes were observed in fusiform and triradiate cells upon treatment? Were those the reasons for lower transformation efficiency?

This is an excellent suggestion, we have added text and microscopy images for cells treated with 0.1, 1, and 10 mAnson alcalase per 10⁸ cells (corresponds to exp. in Fig. 1C) to demonstrate the morphological changes induced by the various alcalase treatments (Supplementary Fig. S3). We very rarely see triradiate forms in healthy wild-type cultures so we couldn't capture images of this particular morphotype (though we have seen triradiate frustules following spheroplasting treatments), but both the fusiform and oval morphotypes can be visibly protoplasted by alcalase. This is best evidenced by the appearance of fusiform and oval 'shells' (i.e., empty frustules) in the 1 and 10 mAnson treated cells – these 'shells' become increasingly more visible as the strength of the alcalase treatment increases. We believe the lower transformation efficiency with liquid grown cultures has more to do with other phenotypic changes that accompany growth on solid media. It's worth noting that a similar phenomenon occurs during conjugation; plated *P. tricornutum* cells are more efficient than liquid cultured cells (Karas et al, 2015; doi: 10.1038/ncomms7925).

Figure 2e:

Please comment on the absence of a band at around 486 bp in lanes 3, 4, and 5 in the text. Could the integration of the introduced DNA fragments into the genome result in resistance to the selection antibiotics?

We thank the reviewer for this insightful question. The absence of the 486 bp band in lanes 3, 4, and 5 is most likely due to DNA degradation or deletions occurring at the episomal termini, either before (e.g., exonuclease activity) or during the non-homologous repair process. While integration of the introduced DNA fragments into the genome could in principle confer resistance, the presence of the other amplicon suggests that the construct is being maintained episomally. This amplicon is located in the CEN/ARS region, which is ~3 kb away from the other amplicon (i.e., the site of termini fusion) and ~7.5 kb away from the *nat* cassette in the linear construct. We have added text to this section to address this.

Page 16, Line 501–:

To ensure the purpose of this experiment is immediately clear to the readers, it would be helpful to briefly state the purpose of the co-delivery of RNPs and episomes in this paragraph.

This is an excellent suggestion, we have added text to the revised manuscript to clearly state the purpose of these co-delivery experiments. Specifically we added this sentence:

"The aim of this experiment was to develop protocol that could significantly improve screening for gene deletions that do not produce an obvious phenotype."

Figure 7:

Although Figure 7 suggests the potential for the targeted insertion of foreign DNA at arbitrary loci in the diatom genome, this possibility has not been demonstrated

experimentally in the current study. Although this is a plausible and exciting prospect, the figure should be omitted to avoid misleading readers.

We agree with this statement; the inclusion of Fig. 7 is too pre-emptive for this manuscript. The figure has been removed and the text has been revised accordingly.

We thank the Reviewer #1 again for their thoughtful comments and for helping us improve the manuscript.

Reviewer #2 (Remarks to the Author):

General Comments:

This manuscript presents promising improvements in DNA delivery protocols for *P. tricornutum*, particularly through the use of spheroplasting in combination with electroporation and polyethylene glycol (PEG)-mediated transformation. The authors demonstrate improved transformation efficiency using both circular and linearized plasmids, and successfully apply ribonucleoprotein (RNP) complexes of Cas9 and gRNA for gene editing. Notably, they provide the first, exciting evidence of in vivo DNA fragment assembly in diatoms.

However, in its current form, the manuscript primarily presents an optimization of existing transformation protocols. While the authors demonstrate successful delivery of various constructs into diatom cells, they do not investigate the fate of the introduced DNA within the algal cells—specifically, whether integration into the genome occurs, the copy number and stability of episomal plasmids, or the potential for epigenetic silencing. Additionally, the novel and potentially impactful observation of in vivo DNA assembly in algae is acknowledged but not explored in depth.

As such, the manuscript may be more appropriate for a technically oriented journal. Alternatively, it could be significantly strengthened by expanding on the mechanisms and biological implications of in vivo DNA assembly in diatoms.

We sincerely thank Reviewers #2 and #3 for their combined extensive feedback, it has helped strengthen and refine our work. We will provide more detailed responses below to the specific comments and suggestions, but first, we wanted to address some of the comments above, starting with:

“While the authors demonstrate successful delivery of various constructs into diatom cells, they do not investigate the fate of the introduced DNA within the algal cells—specifically, whether integration into the genome occurs, the copy number and stability of episomal plasmids, or the potential for epigenetic silencing.”

This is a valid and important criticism, and we have taken steps to address each aspect as follows:

Genomic integration: We have added additional section in the revised manuscript to address whether genomic integration occurs (see also Reviewer #1’s comments). In brief, while genomic integration of the episome can occur, it is probabilistically far less likely than episomal maintenance. Successful integration would require multiple chance events: the episome must first reach the nucleus, and then must integrate into the genome at a sporadic double-strand break (DSB) site located coincidentally within a transcriptionally active genomic region; conversely, for episomal maintenance, the episome just has to

reach the nucleus and circularize, as our work has revealed, for successful transformation. An example of this can be seen in Supplementary Fig. S1, where we transformed equimolar amounts of the episome pPtGE31_ΔPtR (~11 kb) and the nourseothricin resistance cassette (~1.4 kb) into *P. tricornutum*. Despite being nearly tenfold smaller (= higher transformation efficiency), the cassette yielded over tenfold fewer colonies than the episome, underscoring the lower likelihood of successful genomic integration due to the need for a DSB in a transcriptionally active region.

Copy number and stability: We did not explicitly investigate episomal copy number or stability following electroporation or PEG-mediated transformation, as we do not anticipate significant differences compared with other DNA delivery methods – particularly in comparison to biolistics, which also relies on physical force to introduce DNA into cells. The episomes used in this work have been previously characterized and shown to be stable (Slattery et al., 2018; doi: 10.1021/acssynbio.7b00191), and past work has also shown that episomal DNA in *P. tricornutum* is maintained at low copy number within the nucleus (Karas et al., 2015; doi: 10.1038/ncomms7925). We did report on the existence of subpopulations of plasmids derived from a single algal colony, which is consistent with what has been reported for conjugation-based delivery systems (see Diaz-Garza et al., 2024; doi: 10.1186/s12934-024-02559-y).

Epigenetic silencing: We did not characterize the epigenetic fate of introduced episomes as it lies beyond the scope of our work. Furthermore, recent work from the Desgagné-Penix lab has examined the epigenetic fate of pCC1BAC-based episomes (same as in our studies) introduced into *P. tricornutum* in detail (see Diamond et al, 2023; doi: 10.1016/j.algal.2023.102998 and Diaz-Garza et al., 2024; doi: 10.1186/s12934-024-02559-y).

Now, to the second comment:

“Additionally, the novel and potentially impactful observation of in vivo DNA assembly in algae is acknowledged but not explored in depth.”

This is again a very valid criticism, and we performed additional work investigating potential applications for DIVA as well as characterizing the parameters necessary for successful assembly. Specifically, we have devised a system for reliably assembling any gene of interest into an episome directly in *P. tricornutum* and we also characterized that overlaps as small as 20 bp are sufficient for homology-directed assembly. We will delve into further details regarding this below, but want to thank you for this feedback as it has strengthened our work and opened endless avenues for exploration in the lab.

Below are specific comments and suggestions for revision:

Major comments:

Overstatements

The authors claim that DNA delivery is the major challenge in diatom transformation. While this is true for several diatom species, it does not hold for *Phaeodactylum tricornutum*, which can be efficiently transformed using multiple established methods. In contrast, a more significant challenge in *P. tricornutum* is the frequent epigenetic silencing of transgenes. Furthermore, *P. tricornutum* lacks a silica cell wall, making it particularly well-suited for spheroplasting-based approaches.

We can see that the statement “DNA delivery is the major challenge...” was overstated without proper context, and we have revised the text accordingly to better clarify the specific challenges associated with DNA delivery in *P. tricornutum*.

Our rationale for including this statement initially was that, prior to this work, the only well-established and widely used methods for transforming *P. tricornutum* were biolistics and conjugation. Although electroporation-based methods have been described since the early 2010s, they were not widely adopted due to limited efficiency. Biolistics, while functional, typically yields relatively few transformants and is not well suited for delivering large constructs. Conjugation is more efficient but can have other problems. In our own experience, some transmissible plasmids have proven incompatible with the canonical Mob-based helper plasmids, necessitating extensive troubleshooting to achieve successful conjugation (this is the focus of our recent preprint – Walker et al., doi: 10.1101/2025.10.16.682956). Moreover, some transgenes such as endonucleases (e.g., CRISPR-Cas9, Cys4) can be toxic in *E. coli*, causing issues with plasmid stability prior to performing conjugation.

By optimizing the electroporation method ~200-fold and developing a PEG-based transformation protocol, we substantially expand the molecular toolbox for *P. tricornutum* and make this organism more accessible for genetic engineering.

We touched on this in our response above, but we did not investigate the epigenetic fate of introduced episomes as this went beyond the scope of the work, which was focussed on exploring the limits of electroporation and PEG transformation methods. Recent work from the Desgagné-Penix lab has investigated this using pCC1BAC-based episomes (mentioned in comment above).

But we agree that with the reviewer and we adjusted the text both in abstract and introduction where were made such statements in our initial submission.

The inclusion of *Thalassiosira pseudonana* is also unconvincing. As the other main model diatom species, *T. pseudonana* has a well-documented history of successful transformations, and it is unclear how the current protocol represents a major advancement for other species.

We appreciate the reviewer’s comment and the opportunity to clarify our rationale for including *Thalassiosira pseudonana* in this study. As noted in your comment above, *P. tricornutum* has a relatively poorly silicified cell wall, which may make it more amenable to spheroplasting/protoplasting transformation methods. We therefore sought to test whether our approach could also be applied to a more conventional diatom species with a siliceous frustule.

We selected *T. pseudonana* because it already has well-characterized episomal systems and selectable markers, making it an ideal candidate for a proof-of-concept demonstration. While *T. pseudonana* can indeed be transformed using established biolistic and conjugation methods, the PEG-based approach we developed is considerably simpler and faster to perform. As with *P. tricornutum*, this method expands the available molecular toolbox and demonstrates that the protoplasting-based strategy can be extended to diatoms with more heavily silicified cell walls.

The authors further propose that de novo diatom in vivo assembly (DIVA) could replace

traditional cloning approaches in *E. coli* and *S. cerevisiae*, which they describe as time-consuming. However, this claim is not well substantiated. The data presented show that *in vivo* assembly in diatoms is prone to errors, in contrast to the reliable transformation with pre-assembled plasmids. Moreover, assembled episomes still require purification and transformation into *E. coli* for verification. This statement also does not account for the lengthy selection process in diatoms, which typically takes around three weeks, compared to the single-day turnaround in *E. coli*. Instead of positioning DIVA as a general replacement for bacterial or yeast cloning, we suggest that the authors propose a more practical and specific application where DIVA could offer a clear advantage.

We thank the Reviewers for this thoughtful comment. We agree that our original statement – that DIVA could broadly replace traditional cloning in *E. coli* or *S. cerevisiae* – was overly general and not sufficiently supported by the data presented. As the Reviewers noted, *in vivo* assembly in *P. tricornutum* is more error-prone than cloning in bacteria or yeast, and assembled episomes must still be recovered and validated in *E. coli*. The longer selection time in diatoms further limits the feasibility of positioning DIVA as an alternative for the standard molecular cloning workflows.

In accordance with this comment, we performed additional experiments to better characterize both the limitations and potential applications of DIVA. To assess the minimal homology requirements, we performed assembly using a NAT ORF amplified with 0, 20, or 40 bp of overlap to the target episome; these experiments showed that 20 bp is sufficient to direct HDR, consistent with observations in *S. cerevisiae*. This result is practically useful as it means any gene or fragment of interest can be PCR-amplified with primers that add ≥ 20 bp of overlaps and then directly used for DIVA. Previously, we relied on ~ 160 bp overlaps, which cannot be readily generated by PCR.

We next evaluated whether DIVA could be used to test putative localization signals. We fused a predicted mitochondrial signal peptide from the nuclear-encoded, mitochondrially localized protein CytB to eGFP and cotransformed this construct with an episome designed for insertion. Mitochondrial fluorescence was observed in 4 of 10 screened transformants, supporting the functionality of the predicted CytB transit peptide.

The CytB-eGFP assembly was somewhat inefficient in that the episome could circularize through NHR, leading to nourseothricin resistance without the integration of the construct. To address this, we synthesized another gBlock comprising eGFP linked to an incomplete NAT ORF via the previously characterized T2A linker. This design yielded 10/10 transformants with cytosolic eGFP expression upon screening, demonstrating that this strategy provides a reliable approach for direct insertion and expression of a gene of interest via DIVA.

At present, DIVA is viewed as a promising assembly capability rather than a fully optimized platform. We established the basic parameters (e.g., overlap length, number of fragments, relative efficiencies) and demonstrate two proof-of-concept applications. We anticipate that the clearest advantages may emerge in contexts where rapid or pooled assembly directly in *P. tricornutum* is useful – such as screening promoter libraries or constructing variant pools for phenotype-based enrichment – but exploring these applications lies beyond the scope of the present work. We have updated the Results section (*Exploring the limits and applications of DIVA*), added additional data (Supplementary Figs. 11 and 12), and have revised the Abstract/Introduction/Discussion sections to present a more modest and balanced interpretation of DIVA.

Co-transformation with RNP Cas9-gRNA complexes and episomal resistance marker

The manuscript presents this strategy as a solution for targeting genomic loci that lack a selectable phenotype. However, with a reported co-transformation efficiency of only 10%, the majority of resulting colonies (~90%) are false positives. The authors might consider exploring an alternative strategy—such as co-delivery of multiple RNP complexes targeting different loci—similar to the approach described in Serif et al., 2018.

We thank the Reviewers for this thoughtful comment and have added further clarification in the revised manuscript to make the purpose of this experiment clearer (also in accordance with comments made by Reviewer 1). Our aim was to quantify the probability of co-delivering RNPs and episomal DNA into the same cell, facilitating future screens where CRISPR-induced genomic changes do not show an obvious phenotype/cannot be selected for directly. Below we have an illustration to try to visualize what we were trying to quantify (Rudimentary Fig. 1); in short, a cell that takes up episomal DNA has some probability of also taking up an RNP when transformed simultaneously.

Determining the co-delivery rate provides a useful benchmark for experiments with RNPs targeting non-selectable loci as it allows us to estimate the number of colonies that should be screened to identify edits. In this context, the ~10% co-transformation rate tells us that if we co-transform a selectable episome with an RNP targeting a non-selectable locus, we should aim to screen approximately 10 colonies to identify 1 putative RNP-transformed colony. Without this, we have no capacity to select for RNP-transformed cells when the gene being targeted doesn't produce a selectable phenotype – a single transformation uses $\sim 1 \times 10^8$ cells, after all. The 90% of transformants that didn't demonstrate co-delivery of both the RNP/episome are not false positives in this case either; they are simply colonies that were only transformed by one complex. We believe that our demonstrated approach will be highly useful for generating targeted deletions, including the potential creation of a comprehensive deletion library of all non-essential genes.

Rudimentary Figure 1. When simultaneously electroporating an episome (i.e., DNA) and RNP into *P. tricornutum*, the vast majority of cells (i.e., >99.9%) will remain untransformed. Of the transformed cells, some will have taken up only the episome (red hemisphere) or the RNP (blue hemisphere), with some proportion having taken up both constructs (purple intersection). We sought to characterize the percentage of transformed cells that took up both the episome and RNP simultaneously to quantify the probability of co-transformation.

Examples of overstatements:

Line 28: The sentence overgeneralizes the adaptability of the protocol. *Thalassiosira pseudonana* is also a model diatom species with a small genome, rapid growth, and generally accessible for genetic transformation. Given the differences between species and that electroporation was only tested in *P. tricornutum* (and most effective on plates), caution should be used when implying broad adaptability.

We thank the Reviewers for this helpful comment and have revised the text to use more cautious language regarding the adaptability of these methods. We had previously mentioned in the text that *P. tricornutum* is an atypical diatom with a relatively poorly silicified cell wall, which may make these transformation techniques more amenable compared to other species. We want to note that similar concerns were raised when the bacterial conjugation method was first established for *P. tricornutum* in 2015. Since then, conjugation – which also relies on direct DNA transfer – has been successfully adapted for a range of diatom species, including *T. pseudonana* (as reported in the original 2015 study), *Fistulifera solaris*, *Nitzschia captiva*, *Cyclotella meneghiniana*, and other species that have yet to be published (anecdotal; based upon discussions during the MLD 8 meeting). We have added this context to the Discussion to support the rationale for potential adaptability of these methods. The text we added to the Discussion is as follows:

*“These results suggest that the spheroplasting and protoplasting techniques can be applied to other diatoms, though it will likely be necessary to optimize the intensity and duration of the enzyme treatment on a species-by-species basis along with other parameters in the respective protocols. This expectation aligns with the trajectory of bacterial conjugation in diatoms, which was originally established using plated *P. tricornutum* cells: although initially assumed to have limited applicability, conjugation has since been successfully adapted to multiple species, including *Cyclotella meneghiniana*, *Nitzschia captiva*, and *Fistulifera solaris*. This precedent underscores the realistic potential for broader adaptability of the methods presented here.”*

Line 64: “Conjugation requires the creation and maintenance of a donor strain of *Escherichia coli*, which must contain a suitable broad-host-range conjugative plasmid in addition to the episome of interest.” While this statement is technically accurate, it is unclear why the authors present it as a drawback. The creation and maintenance of donor *E. coli* strains is a standard molecular biology procedure and typically not considered a major limitation.

We thank the Reviewers for this observation and have changed the text provide greater context for this statement. While creating donor *E. coli* strains is indeed a standard molecular biology procedure, in practice it can present several unexpected challenges. As noted above, we and others have encountered issues with incompatibility between certain transmissible episomes and the MOB-based conjugative plasmids used for *P. tricornutum* and *T. pseudonana* (see our preprint – Walker et al., doi: 10.1101/2025.10.16.682956). Furthermore, co-maintaining MOB plasmids with medium- or high-copy transmissible plasmids can lead to rearrangements and plasmid instability, particularly when the transmissible construct expresses genes that are considered toxic to *E. coli* (e.g., endonucleases). Thus, when creating a conjugative *E. coli* strain, it’s best practice to reisolate and sequence the plasmid of interest after several passages to validate its stability and integrity over time. In contrast, direct transformation through electroporation or PEG transformation circumvents this step and can therefore shorten experimental timelines by several days to weeks (and potentially even months, as was the case for our

plasmid incompatibility issue). Our intent here is to provide efficient alternative for DNA delivery.

We have expanded the text in the Introduction section to elaborate on these practical drawbacks. The improved text (in the Introduction section) now reads:

“Conjugation requires the creation and maintenance of a donor strain of Escherichia coli, which must contain a suitable broad-host-range conjugative plasmid in addition to the episome of interest; complications can arise if these two constructs have incompatible replication machinery, if the episome is maintained at medium to high copy number, or if it contains elements that are toxic when expressed in E. coli (e.g., endonucleases). These issues can lead to episomal rearrangements or loss over time. As a result, donor strains often require repeated reisolation and plasmid sequencing to confirm construct stability – steps that can add days to weeks, and in some cases months, to a workflow.”

Line 89: Reiteration of line 28 comment: the claim that these methods can be broadly applied to other diatoms should be more cautious and supported with data or omitted.

We agree with this statement and have revised the text accordingly, as stated under the earlier comment regarding Line 28.

General note: The data suggest that spheroplasting improves transformation mainly on plates, with limited benefit in liquid culture. This is important because many diatom species cannot grow on solid media.

We thank the Reviewers for this important point. In our Discussion, we had previously stated that **“*Though P. tricornutum is unique in both its poorly silicified cell wall and willingness to grow on solid media, we were able to adapt the alcalase-based protoplasting technique for T. pseudonana...*”** to address this. The conjugation-based method, which also relies on plated cells for efficient *P. tricornutum* transformation, has been successfully adopted in several other diatom species, suggesting that plating requirements are unlikely to represent a major barrier to broader adoption of electroporation or PEG-based approaches. This ties into our response for the Line 28 comment above. We added the following text to better highlight our earlier statement:

“Though P. tricornutum is unique in both its poorly silicified cell wall and willingness to grow on solid media, we were able to adapt the alcalase-based protoplasting technique for T. pseudonana, a diatom species with a silicified cell wall that is typically cultured in liquid media.”

Uneven quality across the manuscript:

The final two sections—PEG transformation and *Thalassiosira pseudonana* transformation—are notably underdeveloped compared to the rest of the experimental work. These sections appear rushed, lacking sufficient detail, proper quantification, and clear presentation of results. To maintain consistency and clarity, the authors should either expand and revise these sections to match the depth and quality of the earlier parts of the manuscript or consider omitting them altogether.

We thank the reviewers for this valuable suggestion and have expanded both sections accordingly. The fluorescence microscopy of *T. pseudonana* PEG transformants was repeated using proper controls and multiple channels to provide higher-quality and more conclusive data (Supplementary Fig. S17), and the *P. tricornutum* PEG section has been

revised to include additional methodological detail in the methods section. While the PEG transformation was intended as a complementary proof-of-concept rather than a major focus of the study, we agree that the additional data and clarification improve the overall consistency and completeness of the manuscript.

Open questions:

Would the addition of telomeric sequences help stabilize the linearized plasmids and prevent their circularization in diatom cells?

This is an excellent question, and coincidentally, this is (at least in part) where our explorations began. We want to one day install a synthetic linear chromosome in the *P. tricornutum* nucleus as a part of the Pt syn 1.0 project (see Pampuch/Walker et al.; doi: 10.1016/j.cogsc.2022.100611). As a first attempt at this, we PCR amplified the episome pPtGE31_ΔPtR using primers that would add 0, 1, 2, or 5 telomere repeats (AACCTT) to the termini. This was prior to the optimization of the electroporation method, so for these transformations, we generated hundreds of transformants for each of the episomes tested (Rudimentary Fig. 2). Colonies from the 1T, 2T, and 5T transformation plates were picked and screened using a PCR primer set that would only generate an amplicon if the episome had circularized (sparing any major deletions around the episomal termini). For 14 out of the 15 colonies screened, an amplicon was generated – though the size of the amplicons varied, which suggested to us that circularization was occurring, and indels were accruing during circularization (Rudimentary Fig. 2). A few of the episomes were then recovered in *E. coli*, validating that circularization had indeed occurred (discussed further in response to Reviewer #1, see Supplementary Fig. S2). We also sequenced four episomes, confirming that the construct's termini had been ligated through a non-homologous repair process.

Rudimentary Figure 2. Transformation of linear pPtGE31_ΔPtR with 0, 1, 2, or 5 telomere repeats at its termini. (a) pPtGE31_ΔPtR (~11 kb) was amplified using primers that would add up to 5 telomere repeats at its termini. These constructs were transformed alongside the nourseothricin resistance marker (NAT) as a positive control, as up until this point, we had not yet explored the delivery of episomes through electroporation (i.e., we had only explored genomic integration of markers). (b) An agarose gel depicting the amplicons generated from the constructs of panel a. An off-target band appears for the 5-

telomere (5T) construct around the 2 kb mark. (c) Colonies post electroporation of the NAT marker and various pPtGE31_ΔPtR telomere constructs. The negative control consisted of cells electroporated without episomal or integrative DNA.

Rudimentary Figure 3. Screening five transformants from the 1T, 2T, and 5T colonies shown in Rudimentary Fig. 2. DNA was isolated from the colonies and then subjected to PCR screening using a primer set spanning the junction termini. If the episomal ends were reattached seamlessly, we would expect a band of 267 bp (as demonstrated by the positive control, which was circular pPtGE31_ΔPtR). Any insertions or deletions could impact the size or appearance of this amplicon. This screen was conducted using the SuperPlex kit (Takara; 35 cycles), with the negative control consisting of ddH₂O.

Could the authors clarify why adenine supplementation is necessary following PtAPT knockout? What is the underlying metabolic or genetic rationale?

We appreciate the Reviewers' question and have expanded the relevant section accordingly. As noted, this experiment was intended as a proof-of-concept based on the work of Serif et al. (doi: s41467-018-06378-9), who characterized the APT/2-FA counterselection system in *P. tricornutum* in detail. To our knowledge, PtAPT functions in adenine salvage rather than as an auxotrophic gene, and therefore adenine supplementation is not strictly required for growth of PtAPT-knockout strains under normal conditions. However, we observed that adenine supplementation substantially improves recovery of transformants following electroporation of the Cas9:PtAPT complex. When plated on 2-FA selection medium, plates lacking supplemental adenine produced only a handful colonies, whereas plates with supplemental adenine yielded thousands (Supplementary Fig. S14).

This finding arose somewhat serendipitously. While optimizing 2-FA selection parameters in our hands (simply because different batches/manufacturers of 2-FA can vary in potency), wild-type *P. tricornutum* cells plated on media containing 5 mg/L adenine but no 2-FA showed unexpectedly poor growth. Subsequent tests confirmed that adenine supplementation of 1 to 5 mg/L was surprisingly toxic to wild-type cells (Supplementary Fig. S13). Given this, we compared plating of Cas9:PtAPT transformants on 2-FA medium with and without adenine to assess whether supplementation would affect recovery, hypothesizing that adenine might also stress PtAPT-knockout strains. Notably, Serif et al reported plating Cas9:PtAPT transformants on 2-FA plates supplemented with 5 mg/L adenine; we are uncertain why adenine was added in their experiments, it was not fully

clarified there either. In our experiments, as mentioned above, adenine supplementation proved to make a significant difference in the recovery of transformants (though our initial hypothesis was incorrect).

Mechanistically, APT catalyzes the transfer of phosphoribosyl pyrophosphate (PRPP) to adenine to form AMP. In the presence of 2-fluoroadenine (2-FA), APT instead converts 2-FA into 2-fluoro-AMP, which is subsequently phosphorylated to form toxic fluorinated nucleotides (i.e., 2-fluoro-ADP/ATP). When APT is knocked out, this conversion no longer occurs, preventing toxicity. Based on this, we are uncertain why the addition of adenine improves the recovery of PtAPT transformants so drastically; perhaps there are other adenine salvage/recycling/metabolic pathways, and the presence of supplemental adenine reduces the toxicity of 2-FA in these alternative pathways. This would be our best hypothesis, especially since adenine supplementation is not necessary when passaging the cultures in the absence of 2-FA.

Together, these results suggest that while adenine supplementation is not strictly required for PtAPT-knockout strains, it enhances transformant recovery and acts as a metabolic buffer under 2-FA selection. The adenine toxicity observed in wild-type cultures further implies that purine salvage and PRPP homeostasis are tightly regulated in *P. tricornutum*. Further investigation into this regulation would be valuable but lies beyond the scope of our work; we have added some text to comment on this phenomenon in the Results section and invite further inquiry in the Discussion section. The text reads as follows:

RESULTS

“Serif et al. had previously shown that knock-out of the P. tricornutum Adenine Phosphoribosyl Transferase (PtAPT) gene confers resistance to the cytotoxin 2-fluoroadenine (2-FA) at a dose of 10 μM. PtAPT functions in the adenine salvage pathway, catalysing the transfer of phosphoribosyl pyrophosphate (PRPP) to adenine to form adenosine monophosphate (AMP); in the presence of 2-FA, PtAPT instead converts 2-FA into 2-fluoro-AMP, which is subsequently phosphorylated to form toxic fluorinated nucleotides.

Chemicals can vary based on the manufacturer and/or batch; thus, when setting forth on this series of experiments, we first tested the resistance of wild-type P. tricornutum (CCAP 1055/1) cultured on ½-salt L1 plates supplemented with adenine (5 μg/ml) and our batch of 2-FA (0, 2.5, 5, 10, 15, 20 μM). To our surprise, there was an absence of growth on the 0 μM 2-FA plate, suggesting that adenine alone posed a toxicity to wild-type P. tricornutum cells. We tested this theory by plating wild-type cells onto ½-salt L1 plates supplemented with varying levels of adenine (0, 1, 2.5, 5 μg/ml; Figure S13). The growth of wild-type algae was negligible after one-week on the 5 μg/ml adenine plate, but relatively unimpacted on the 0.5 and 1 μg/ml plates. This interesting revelation led us to hypothesize that adenine supplementation could negatively impact ΔPtAPT transformants as well, so when conducting our RNP electroporation experiments, we plated transformants on 2-FA plates with and without adenine supplementation.

...

Transformants were recovered for one day before plating one-fifth of the reaction across ½ salt L1 plates supplemented with 10 μM or 5 μM 2-FA, with or without 5 μg/ml adenine (Supplemental Fig. S14). This pilot experiment generated thousands of 2-FA resistant transformants across the plates that had been supplemented with adenine, but only a

handful across the plates without supplemental adenine. This was an unexpected discovery given our previous observations. Ten 2-FA resistant colonies were passaged for further screening at the PtAPT loci to determine if Cas9-mediated engineering had occurred.”

DISCUSSION

*“Through this series of proof-of-concept experiments, we also uncovered an unexpected paradox: while adenine supplementation (5 µg/ml) is visibly toxic to wild-type *P. tricornutum*, it substantially improves the recovery of Δ PtAPT transformants under 2-FA selection. Because PtAPT functions in the adenine salvage pathway, it is not immediately clear why supplemental adenine would have any beneficial effect on Δ PtAPT strains; if anything, one might expect the opposite. This result suggests that additional adenine salvage, recycling, or purine-metabolic pathways may be active in *P. tricornutum*, and that supplemental adenine may mitigate 2-FA toxicity through these alternative routes. Elucidating the underlying metabolic regulation will require further investigation, but is beyond the scope of the present study.”*

Additional line-by-line comments:

Line 26: The term “PEG” is used in the Abstract without being defined. Please define “polyethylene glycol” at first mention.

Thank you for catching this, we have corrected the Abstract accordingly.

Line 63: The transformation rate is incorrectly written as “4 transformants per 10^{-4} cells”. It should read “4 transformants per 10^4 cells” for consistency with line 58.

Thank you for catching this, we have corrected this error and all other related errors accordingly.

Line 69: More background and context on PEG-mediated transformation in diatoms is needed. Currently, the discussion is too brief for readers unfamiliar with the technique.

We appreciate this suggestion and have expanded the background on PEG-mediated transformation accordingly. To our knowledge, PEG transformation has only been described once in diatoms (Karas et al., doi: 10.1038/ncomms7925), where the method was reported to be inefficient and inconsistent, yielding only 2–3 transformants across dozens of attempts. In revisiting this approach, we followed the overall workflow of Karas et al. but introduced several key modifications: we used alcalase as the protoplasting enzyme instead of the previously reported zymolyase/hemicellulase/lysozyme cocktail, we included salmon sperm DNA during the transformation step, and we recovered the protoplasted cells in liquid media and for a longer duration before plating. These adjustments substantially improved the robustness and reliability of the protocol, routinely producing thousands of transformants per experiment when transforming pPtGE31_ΔPtR.

We have added the following text to the background section:

*“Although PEG-mediated transformation has not yet been demonstrated in other diatom species, it is routinely applied in the microalga *Chlamydomonas reinhardtii*, where cell-wall removal followed by PEG-mediated agitation with DNA and glass beads enables efficient nuclear transformation.”*

Line 70: Correct “≤ 1 transformant per 10⁻⁸ cells” to “≤ 1 transformant per 10⁸ cells”.

Thank you for catching this, we have corrected this error.

Line 87: The manuscript inconsistently uses the terms spheroplasting and protoplasting. These are sometimes used interchangeably and sometimes as distinct processes. Given the title refers specifically to “spheroblasted cells,” a clear and consistent definition is essential.

We thank the Reviewers for noting this inconsistency. We have reviewed the manuscript to ensure the terminology is used consistently and now clearly define our usage of “spheroplasting” and “protoplasting”. The following text was added to make this distinction clear:

“Notably, a spheroplasted cell has only partial cell wall degradation and may not appear phenotypically different from a wild-type cell, whereas a fully protoplasted cell lacks a cell wall entirely and adopts a characteristically spherical morphology – provided it does not burst from osmotic or mechanical stressors.”

We also propose a change the title of the manuscript to read “Breaking the cell wall: a gamechanger for DNA delivery to diatoms” to hopefully resolve this inconsistency.

Line 97: The authors note that a neighbouring lab used older electroporation parameters and that the switch from 25 to 50 μF may relate to the equipment used. This raises the question of whether the improved transformation efficiency is protocol-specific or simply equipment-dependent. Please clarify.

We appreciate the Reviewer’s point and want to clarify – the substantial increase in transformation efficiency results from the spheroplasting step, not from changes to the electroporation pulse parameters. Prior to optimizing the resistance settings for our specific electroporator, we were unable to obtain transformants using the previously reported parameters (i.e., zero transformants across tens of experiments); after adjusting the resistance to values compatible with our equipment, we achieved baseline transformation efficiencies of ~10-250 colonies per experiment. It was the addition of the spheroplasting step that produced the large, order-of-magnitude improvement reported in the manuscript.

Our note about a neighbouring lab successfully using the original pulse settings was intended only to emphasize that electroporation parameters can be equipment-dependent, and users may need to empirically optimize these settings for their own systems. The underlying boost in efficiency is protocol-specific and reflects the impact of spheroplasting, not differences in electroporators.

Line 101: Plasmid pPtGE31_ΔPtR with ID 236260 is not found on Addgene. Please upload the plasmid map and sequence before any resubmission.

All of our plasmids had been submitted to Addgene at the time of initially submitting this manuscript, they can only be found by typing the ID directly into the link since they are not publicly available yet. These plasmids will be released upon publication of the manuscript.

Here is the corresponding plasmid, you can view any other listed plasmids by swapping the last 6 numbers in the link with the corresponding ID: <https://www.addgene.org/236260/>.

Line 176: The use of 500 ng of a PCR-amplified 11-kb construct should be mentioned earlier in the text, especially when referring to Figure 1.

We agree and have made the corresponding change.

Line 256: The claim that linear pSC5 is three times more efficient than circular is not clearly supported by Figure 3B, where colony counts appear similar (5 vs. 6 CFUs). Please clarify.

We agree and have removed this statement, the colony counts are not substantially different enough to support this claim.

Line 302: Plasmid pPtGE31_ShBle with ID 236261 is not found on Addgene. Please upload the plasmid map and sequence before any resubmission.

See the response to the comment on line 101 – you can find the plasmid here: <https://www.addgene.org/236261/>. It will not be searchable or publicly available until the manuscript is published. Everything has been sequence validated by Addgene and is ready for distribution otherwise.

Deposit Information

 Deposited Plasmids - 0 out of 4 are available

Showing 1 to 4 of 4 entries

Show entries

Search table:

Previous **1** Next

Addgene ID	Plasmid	Depositing Organization	Principal Investigator	Sample Status	Hold for Publication?	Available?
236256	pPtGE30	University of Western Ontario	Karas, Bogumil	Ready to distribute	Yes	No
236257	pPtGE31	University of Western Ontario	Karas, Bogumil	Ready to distribute	Yes	No
236260	pPtGE31_Δ PtR	University of Western Ontario	Karas, Bogumil	Ready to distribute	Yes	No
236261	pPtGE31_S hBle	University of Western Ontario	Karas, Bogumil	Ready to distribute	Yes	No

 Deposit Agreement - Completed

Line 393: There is no Figure S6C. This likely refers to Figure S7C—please correct.

Thank you for catching this, we have corrected this in the text.

Line 539: A brief conclusion summarizing the main findings and potential applications would strengthen the end of the manuscript.

This is an excellent suggestion. This was also asked by first reviewer and we added the following text:

"The aim of this experiment was to develop protocol that could significantly improve screening for gene deletions that do not produce an obvious phenotype."

Line 550: The difference in DNA quantities used (500 ng linear vs. 25 µg circular) is substantial. Please explain how these values were chosen.

When this method was first explored by Karas et al in the 2010s, PEG transformation efficiency was very low (2-3 colonies across several experiments), and it was standard practice to add as much circular plasmid DNA as possible in an attempt to increase the number of transformants. At that time, it was not yet known that linearized episomal DNA transforms more efficiently than circular DNA in diatoms.

In revisiting the protocol for the present study, we intentionally used 500 ng of linear episomal DNA (pPtGE31_ΔPtR) so that the PEG-based transformations could be directly compared to our electroporation experiments, all of which also used 500 ng of the same construct. The fact that we were able to obtain thousands of transformants with only 500 ng of DNA further highlights the substantial improvements in efficiency achieved with the updated PEG protocol.

Line 554: Early in the manuscript, alcalase dosage is reported in mAnson units per 10⁸ cells, but here it is given in µl. Please standardize units throughout.

We thank the Reviewers for noting this inconsistency. We have now added the corresponding mAnson unit values wherever alcalase dosage is reported.

Line 558: The text states that 40 µg of carrier DNA improved electroporation efficiency ~15-fold, but no results are shown for PEG transformation. Was this amount used in PEG protocols? Was a similar improvement observed?

We also tested the addition of carrier DNA in the PEG-based transformations (Supplemental Fig. S15B, condition #1 vs #2) and reported ~10-fold difference, similar to what is seen for electroporation. For this step, the key factor is providing sufficient nonspecific DNA to protect the episomal construct from DNases secreted or released by *P. tricornutum* cells; in our experience, 40 µg is more than adequate (especially when using 500 ng of an episome, ssssDNA is 80-fold more abundant).

Line 565: The final optimized PEG protocol is not clearly presented. Please summarize the final conditions used and their effectiveness.

Thank you for this comment, we have revised the text to provide a more clear description of the PEG protocol.

Line 566: There is no Figure S2D, despite the reference. Please correct or include the figure.

We have corrected this, thank you for catching that.

Lines 565–571: This paragraph lacks quantitative data and clarity. It is difficult to assess the advantage of PEG transformation without specific efficiency data.

We thank the Reviewers for this comment. We did not perform enough biological or technical replicates to generate quantitative data for the PEG approach (i.e., lacking data that can be statistically analysed). The PEG protocol was included in this manuscript to demonstrate a robust, equipment-independent alternative for DNA delivery – particularly for species where electroporation may be challenging because the pulse parameters have yet to be quantified.

Although we do not provide formal efficiency measurements, the PEG method consistently produced readily recoverable numbers of transformants under the optimized conditions described (using 500 ng linear pPtGE31_ΔPtR – 1000s of colonies generated), and thus represents a practical, reproducible alternative approach. The plates in Supplemental Fig. 15C conditions #2 to 5 demonstrate the efficiency, keeping in mind that only ¼ of the transformants are shown on these plates. We have revised the text to clarify the purpose of this section accordingly.

*“A polyethylene glycol (PEG) transformation method for *P. tricornutum* with an efficiency of ≤ 1 transformant per 10^8 cells was previously described by Karas et al. For this method, algal cells were protoplasted using a combination of zymolyase, lysozyme, and hemicellulase ahead of PEG treatment. We revisited this method using alcalase as the sole enzyme for cell wall removal, with the intention of providing a simple and equipment-independent alternative to electroporation that is particularly useful for labs or species where this method remains a practical barrier.”*

Line 574: Reiterate the need to clearly outline the final optimized conditions.

We have resolved this in accordance with the comment on Line 565.

Line 577: Please include the size of the episome used here, as done in the Materials and Methods section.

Thank you for catching this, we have included the size of pBIG1 (8224 bp, listed as ~8.2 kb).

Lines 578–581: The results described are vague. Clarify what “some degree of growth” means and how harsher protoplasting conditions influenced colony formation.

This is a valid point. In the revised manuscript, we now refer specifically to the emergence of colonies across the different alcalase treatments. Because *T. pseudonana* does not grow robustly on solid media, reliable colony counts were difficult to obtain for the various conditions. Although transformed *T. pseudonana* cells do form discrete colonies, we often observed smeared or diffuse growth which made accurate counting challenging. We have revised the text to make this more clear.

Line 588: The phrase “a higher proportion of protoplasted cells are visible...” is vague. Provide more concrete observations or quantification, if available.

This is a valid point as well, this statement was based off of microscopy of the cells following treatment with 10 or 1000 ul of alcalase (a snapshot is presented in Supplemental Fig. S16; proper quantification was not performed). Based on these images, it appears as though ~ 1/28 visible cells are protoplasted in panel **b** (= 10 ul alcalase), and ~ 13/20 are protoplasted in panel **c** (=1000 ul alcalase). We have added this to the text to provide a rough quantification of the extent of protoplasting with different alcalase amounts. We have added the following text to address this:

“Based on these microscopy images, cells from the 10 μ l treatment showed a protoplasting rate of ~4% (1/28 cells), whereas cells from the 1000 μ l treatment demonstrated a protoplasting rate of ~50% (13/20 cells).”

Line 593: There is no discussion on partial vs. complete cell wall removal. Is this dependent on enzyme concentration? Do cells survive full wall removal?

For the PEG method, we observed that higher amounts of alcalase and longer incubation times (i.e., more protoplasted cells) yielded more transformants for both diatom species tested. Fully protoplasted *P. tricornutum* and *T. pseudonana* cells survive complete cell wall removal, but care must be taken during handling, as the cells are more fragile and prone to mechanical rupture. It is also important to maintain cells in full-salt L1 (or F/2) media to prevent bursting due to osmotic pressure. In contrast, for electroporation, partially spheroplasted cells can be susceptible to bursting because they are washed in 375 mM D-sorbitol, which has lower osmolarity than L1; this issue is avoided in the PEG method, as cells are maintained in L1 throughout the procedure. We have made changes to the PEG section (in accordance with prior suggestions) to highlight this and to state that >80% of the *P. tricornutum* cells should be protoplasted before proceeding to the next step.

Line 611: It is true that plated cells yielded the highest transformation rates, but based on Table S2, efficiencies are mostly in the 10^6 – 10^7 range, with only treated plated cells showing higher values (1.19×10^4). The benefit of alcalase treatment seems to apply mainly to plated cells.

This interpretation is correct based on the experiments we conducted; however, we did not pursue extensive optimization of liquid-grown cultures because plate-derived cells were already sufficient for establishing a robust protocol in *P. tricornutum*. Numerous variables remain unexplored in liquid cultures, for example recovery in liquid rather than on plates, or selection conditions better suited to cells not adapted to solid media, time of protoplasting and amount of alcalase. It will be interesting to determine how many additional diatom species benefit from cell wall removal, but we do not believe *P. tricornutum* is unique in this regard.

Line 612: The manuscript states that transformation is “efficient,” but no numerical threshold is provided. Please define what constitutes efficiency in this context.

Thank you for pointing this out, we have added the previously stated efficiency (~6 transformants per 10^6 cells) to this sentence to make it more clear.

Line 645: “Several-fold more efficient” should be quantified.

Thank you for the suggestion; to our understanding, Serif et al. generated 16 2-FA resistant transformants across their two biolistic experiments with RNPs targeting the *PtAPT* locus. We generated >1000 transformants in a single experiment (it was difficult to count given the size of the colonies, but given that we plated 1/5 of the reaction per plate, this number is likely closer to ~5000 per transformation), so we will report this as a >200-fold increase to be conservative.

Line 701: Only PEG transformation was adapted to *T. pseudonana*. The claim that these methods hold great promise for other diatoms lacks sufficient supporting evidence.

As noted in earlier responses, while we only tested PEG transformation in *T. pseudonana*, the successful adaptation of this method to a diatom with a canonical silicified cell wall demonstrates its broader potential. Similar hesitations were raised when bacterial conjugation was first developed for *P. tricornutum*; yet, this approach has been successfully adopted across a diverse range of diatom species. We have revised the text in accordance with an earlier comment (Line 28) to clarify this context and temper claims regarding broad applicability while highlighting the proof-of-concept value of our results.

Line 866: The note about keeping materials on ice is useful and should be moved earlier in the protocol section for clarity.

This is a great suggestion, we have added this step earlier to the protocol in accordance.

Line 880: The DNA input range (“1 to 1000 ng”) is very broad. Consider adding a note that for ~11-kb episomes, 500 ng appears to saturate the reaction.

We agree with this and have added a note to the methods section in accordance.

Line 904: This line again references equipment differences affecting electroporation settings. Consider addressing this earlier in the Methods for better context.

We agree with this and have accordingly positioned this text at the start of this methodology section.

Figure Comments:

Figure 1b: Why is the image shown with 100 mAnson if the best treatment was achieved with 1 mAnson? This is confusing and needs explanation.

We wanted to depict what over spheroplasted/protoplasted cells look like; when the cells are spheroplasted with 1 mAnson, they do not look different from untreated cells. In accordance with suggestions from Reviewer 1, we have added high resolution microscopy images for the 0.1 mAnson, 1 mAnson, and 10 mAnson treatments to further characterize the visible changes that emerge during the different spheroplasting intensities (Supplemental Fig. S1).

Figure 1e and f: Statistical analysis should be included. Please clarify what alcalase treatment was used and provide full experimental conditions (e.g., DNA concentration, treatment duration) so that the figure is self-contained.

Thank you for this comment, we have now performed statistical analyses to demonstrate that alcalase-treated, plated cells show significantly higher electroporation efficiency, as

presented in Fig. 1E. Statistical testing was not applied to Fig. 1F, as this panel was intended to illustrate qualitative trends rather than compare the different DNA concentrations quantitatively. We have also updated the Fig. 1 legend to provide additional details on the experimental conditions for panels E and F

Figure 1e: As noted above, experimental details (e.g., DNA concentration, alcalase protocol) are missing. These are essential for interpretation.

We have addressed this in accordance with the comment above, thank you for catching this.

Figure 1f: Clarify whether this experiment was done in liquid culture or on plates, and specify the alcalase treatment used.

We have addressed this in accordance with the comment above, thank you for catching this.

Figure 2c–e: Please clarify the annotations in panels d and e to clearly indicate which PCR primer pairs were used, with explicit reference to panel c.

Thank you for pointing this out, we have made the changes to Fig. 2 accordingly.

Figure 3e is not clearly explained. Presumably, it should refer to line 269: “conjugation was successful for seven out of eight *E. coli* pools”. If it is the case, then labelling C1-C4 in both cases is confusing.

We have revised Fig. 3 and the corresponding in-text reference to make this clearer, thank you for bringing this to our attention. To us, it makes the most sense to keep the labelling as 1 through 4 as we were comparing two conditions (linear vs circular pSC5); to make this distinction clearer to the reader, however, we relabelled the linear-pSC5-derived colonies with L1 to L4.

Figure 5: Part a shows three schemes of possible assembly. Part b does not correspond to part a. Please adjust for clarity.

Thank you for this helpful comment, we agree that the distinction between parts A and B/C could be made clearer. Our intention in the current version of Fig. 5 was to present the episomal assembly workflow in a way that captures the full process: an overview schematic of the possible assembly configurations, followed by one representative plate series, and then one representative set of screening gels. This mixed presentation allows the reader to see both the experimental outcome (colonies) and the downstream verification (screening PCRs in *P. tricornutum* and *E. coli*), rather than limiting the figure to only plates or only gels.

With that said, we recognize that this layout may not appear directly matched to the schematics in part A. To address this, we have prepared two alternative versions of Fig. 5 – one showing all plate series, and one showing all screening gels – to more closely mirror the structure of panel (a). If the Reviewers feel that either alternative improves the clarity of the Figure/manuscript, we are willing to change it to what they see as best fit.

Rudimentary Figure 4. An alternative presentation for the assembly experiments. Here, all of the screening gels are presented below the assembly schematic.

Rudimentary Figure 5. Another alternative presentation for the assembly experiments. Here, all of the experimental plates are presented below the assembly schematic.

Reviewer #3 (Remarks to the Author):

Thank you for taking the time to thoroughly assess our manuscript, along with Reviewer #2 – your comments have greatly strengthened our work.

RESPONSE TO REVIEWERS

Our responses are in purple.

In addition to the comments left by the reviewers, we identified a mistake in our last submission – we had cited ASAFind as the resource we used to identify the CytB signal peptide, when in fact, we had used TargetP (version 2.0). ASAFind was used in a separate project to identify bipartite chloroplast targeting signals of putative nuclear-encoded chloroplast-localized proteins of *P. tricornutum*. The canonical mitochondrial localization signal in diatoms is unipartite and shares conserved features across eukaryotes; thus, it can be identified using TargetP or SignalP. This has now been corrected in the manuscript.

We have also made changes in accordance with the NCOMMS Author Checklist (e.g., refining the abstract, reordering of manuscript sections, changes to word choices).

Reviewer #1 (Remarks to the Author):

The authors have addressed my peer review comments appropriately. I have no further points to raise.

We thank Reviewer 1 for their constructive comments during the initial review and are pleased that the revised manuscript has addressed their concerns.

Reviewer #2 (Remarks to the Author):

We would like to thank the authors for carefully addressing our previous comments. The manuscript is now substantially improved, and we particularly appreciate the more in-depth analysis of in vivo assembly in diatoms, which opens exciting new avenues for future research. While we have one major comment and a few minor remarks, we consider this work a valuable resource for the diatom research community and believe it deserves publication in Nature Communications.

We thank Reviewers 2 and 3 for their careful re-evaluation of the revised manuscript and for their encouraging assessment. We will address the remaining major and minor comments below.

Main comment

While the work on *Phaeodactylum tricornutum* is very convincing, the transformation of *Thalassiosira pseudonana* still raises some questions.

The circular episome pBIG1 (8.2 kb) conferring nourseothricin resistance and carrying eGFP was used in these experiments.

– Is the sequence of this plasmid available or will it be made available?

Yes, the plasmid sequence will be made publicly available. This plasmid is also used in a separate manuscript that is currently under review (Mock lab), and we plan to release the full sequence upon finalization of that work. It will be deposited to Addgene as we have done in the past. Please see link for our lab materials deposited at Addgene: https://www.addgene.org/Thomas_Mock/.

– Which promoter drives the expression?

The promoter is FCP (fucoxanthin-chlorophyll *a/c* binding protein), which is a common promoter used in diatom *Thalassiosira pseudonana*.

– Does it express only eGFP, or are there other expression elements present?

The plasmid has also a *T. pseudonana* gene of unknown function (NCBI ID: XP_002293646) as this plasmid is being used for more than one project. However, this does not impact the expression of eGFP, which is the purpose of this study.

Although in most cases the alcalase units were standardized to mAnson, in this part off the manuscript, alcalase is still in μ l alcalase

Thank you for catching this, we have standardized the alcalase units in this section and in Supplementary Fig. S16 to be consistent with the rest of the manuscript.

In Supplementary Fig. S17, the statement “Both autofluorescence and eGFP signals were visible in the pooled transformed cells, demonstrating that episome transfer had occurred” is difficult to evaluate. The figure is challenging to interpret: although there is no GFP signal in the untransformed control, the green signal observed in the transformed sample appears in structures lacking chloroplast autofluorescence and not

corresponding to *T. pseudonana* cells. This raises concerns about what is being visualized and whether the image may be overexposed.

Hope we can address these concerns by explaining how microscope images of *T. pseudonana* are being influenced by the morphology of the cells. Their morphology is very different to *P. tricornutum* and if the reviewer is not so much familiar with *T. pseudonana*, it is likely that images are being misinterpreted.

Morphology: Generally, *T. pseudonana* cells under the microscope have two different optical axes according to their orientation (Rudimentary Fig. 1). One is called valve view, and the other one is called girdle-band view. If the cells are imaged in valve view, they appear as round discs as shown in our paper. Those round discs might be misinterpreted as not being cells of *T. pseudonana*. However, they are the same cells but just in valve view.

Rudimentary Figure 1. *Thalassiosira pseudonana* under transmission electron microscope. (<https://asknature.org/strategy/protein-directs-silica-growth/attachment/thalassiosira-pseudonana/>)

Chlorophyll autofluorescence: If cells are imaged in valve view, it is much more challenging to obtain a strong chlorophyll signal because of the cellular content pilling

up vertically and if chloroplasts are at the bottom of the pile, the signal of chlorophyll autofluorescence will be very weak (depending on orientation and focal depth), which is what we see sometimes in the image we have added to the paper. Thus, weak autofluorescence does not indicate absence of the cell especially because the cell outline was confirmed by the brightfield channel of the microscope.

Exposure / saturation control: The fluorescence images were not overexposed during acquisition. We ensured that camera settings avoided signal clipping (no saturated pixels) and that the transformed and untransformed samples were imaged using the same acquisition settings. To further alleviate concern, we provide an additional representative field of view (same strain, same imaging session) in which the cell morphology and the two fluorescence channels are clearer (Rudimentary Fig. 2).

Rudimentary Figure 2 *T. pseudonana* transformed cells under fluorescence microscope. A) for bright field, B) for merge, C) for chloroplast autofluorescence and D) for eGFP signal.

Minor remarks

- Although the authors have corrected most instances of “ μ l” to proper enzyme activity units for alcalase, a few remain, mainly in the *T. pseudonana* section.

Thank you for catching this, we have gone through the manuscript again and have hopefully caught all remaining instances of this. In many sections, we report both the enzyme activity (Anson units) and the corresponding volume (μ l). Our reasoning for doing so is to illustrate how ‘potent’ the enzyme is/how little was needed for

spheroplasting in the electroporation section (1 μ l of enzyme per $\sim 3 \times 10^8$ cells!). If preferred, we are happy to standardize all reporting to Anson units alone.

- Line 80: “Electroporation has a higher transformation efficiency (2.6 to 4.5 transformants per 10^5 cells^{5,7})” → Please standardize the citation format with line 62, either inside or outside the parentheses.

Thank you for catching this, we have placed the citations outside of the bracket and have scanned the manuscript for any other instances where we inconsistently formatted the citations.

- Line 354: The correct reference is Supplementary Table S6, not S5.

Thank you for catching this, we have corrected the Table reference accordingly.

- Line 635: “develop a protocol” → missing the article “a”.

Thank you for catching this, we have corrected the typo in the manuscript.

- Keep the rudimentary Figure 5 with the plates in the main paper, and move the figure with the gels to the Supplementary Information.

We agree with this suggestion and have made the according changes to the manuscript and supplementary file. We have also updated the in-text references to correspond with these changes accordingly.

Side remark (for future consideration):

It is possible that the 5× telomeric repeat used is not sufficient to establish a fully protective protein complex equivalent to the human shelterin complex. For future experiments, the authors might consider testing longer telomeric repeats, although these can be difficult to clone and maintain in bacteria.

Yes, this is a great point. We previously did a bioinformatic analysis of the characterized telomeres in *P. tricornutum* and saw that there are as few as 11 repeats naturally, which is remarkably ‘short’ compared to human telomeres. Still, it will likely be necessary to increase telomere/subtelomere length beyond a mere 5 repeats for stable long-term maintenance. Another hypothesis that we are contending with is that our initial construct

may have simply been too small (~12 kb). Previous work done in *S. cerevisiae* demonstrated that linear chromosomes smaller than 100 kb cannot be stably replicated unless circularized – presumably due to issues with replication and/or segregation. We have work underway to try and characterize these parameters as our ultimate goal is to establish synthetic chromosomes in *P. tricornutum* (as a part of the Pt syn 1.0 project).

Reviewer #3 (Remarks to the Author):
